# Coccolith mass and morphology of different *Emiliania huxleyi* morphotypes: A critical examination using Canary Islands material

Simen Alexander Linge Johnsen[ID]*, Jörg Bollmann

Department of Earth Sciences, University of Toronto, Toronto, Ontario, Canada

* simen.johnsen@mail.utoronto.ca

**Data Availability Statement:** All relevant data are within the manuscript and its Supporting Information files.

## Abstract

Different morphotypes of the abundant marine calcifying algal species *Emiliania huxleyi* are commonly linked to various degrees of *E. huxleyi* calcification, but few studies have been done to validate this assumption. This study investigated therefore whether *E. huxleyi* morphotypes can be related to coccolithophore calcification and coccolith mass. Samples from January (high productivity) and September (low productivity) 1997 at an open ocean and a coastal site near the Canary Islands were analysed using a combination of thickness measurements (Circular Polarizer Retardation estimates (CPR) method), Scanning Electron Microscope imaging, and Markov Chain Monte Carlo (MCMC) models. Mean *E. huxleyi* coccolith mass varied from a maximum of 2.9pg at the open ocean station in January to a minimum of 1.7pg in September at both stations. In contrast, overall calcite produced by *E. huxleyi* (assuming 23 coccoliths/cell) varied from a maximum of 2.6 µgL⁻¹ at the coastal station in January to a minimum of 0.5 µgL⁻¹ in September at the open ocean site. The relative abundance of "Overcalcified" Type A, Type A, Group B and malformed coccoliths was determined from SEM images. The mean coccolith mass of "Overcalcified" Type A was 2.0pg using the CPR-method, while mean mass of Type A and Group B coccoliths was determined using coccolith length measurements from SEM images and MCMC models relating thickness measurements to morphotype relative abundance. Type A cocccolith mass varied from a 1.6pg to 2.6pg and Group B coccolith mass varied from 1.5pg to 2.0pg. These results demonstrate that the coccolith mass of Type A, "Overcalcified" Type A, and Group B do not differ systematically and there is no systematic relationship between relative abundance of a morphotype and the overall calcite production of *E. huxleyi*. Therefore, morphotype appearance and relative abundance can not be uniformly used as reliable indicators of *E. huxleyi* calcification or calcite production.

## Introduction

Since the industrial revolution, anthropogenic $CO_2$ release into the atmosphere has led to a ~0.1 decrease in surface ocean pH in a process called ocean acidification [1]. While $CO_2$

**Funding:** This work was funded by NSERC through J. B.'s NSERC Discovery Grant [https://www.nserc-crsng.gc.ca/] and by the Canadian Network for Research and Innovation in Machining Technology, Natural Sciences and Engineering Research Council of Canada. Sampling was funded by the E. U. 598 project CANIGO. The funders had no role in study design, data collection and analysis, decision to publish, or preparation of the manuscript.

**Competing interests:** The authors have declared that no competing interests exist.

continues to increase in the atmosphere and surface ocean, pH may drop by another 0.4 units by the end of the century [2] and the $CaCO_3$ saturation state of the ocean may drop by 2.0 units [3]. Because of the drop in pH and $CaCO_3$ saturation state, calcifying organisms are thought to be particularly affected by ocean acidification [2–4], including coccolithophores [5–8].

Coccolithophores are an abundant group of marine algae that play an important part in the marine carbon cycle as both primary and calcite producers [9]. Much research has therefore focused on the effects of ocean acidification on calcification in coccolithophores, particularly the species *Emiliania huxleyi* (e.g. [5–7, 10–14]). The research has focused on *E. huxleyi* because it is the most abundant coccolithophore species in the modern ocean [9, 15]. In addition, *E. huxleyi* is relatively easy to culture in a laboratory [9]. However, studies on calcification in *E. huxleyi* are non-conclusive, with different studies reporting decreased (e.g. [5, 6]), increased (e.g. [10, 12]), or unchanged (e.g. [13, 14]) calcification with decreasing pH. The non-conclusive results may be due to methodological issues [16, 17], or strain-specific responses (e.g. [11]), compounded by the non-standardised use of the term calcification. Calcification is a rather loosely used term which encompasses the entire calcite production process from the transport of ions to precipitation of calcite [18]. The amount of calcite produced per cell (often in relation to organic carbon content, the PIC/POC ratio) or the calcite production rate are usually the specific parameters of interest in laboratory studies (e.g. [5, 10]). These parameters can readily be measured in cultured samples using various established methodologies [19–23], but these methodologies are not appropriate for species-specific data from field studies where numerous different calcitic organisms are present. Several studies have therefore relied on the weight/mass estimation of single coccoliths as a proxy for coccolithophore calcification (e.g. [8, 24, 25]). Different methods have been used for estimating coccolith mass, including the volumetric model of [26] based on coccolith length measurements (e.g. [27, 28]), mass measurements based on interference colour (e.g. [8, 29]), measurements of central tube thickness (e.g. [30, 31]), and the qualitative analysis of coccolith morphology from SEM images (e.g. [32, 33]).

The latter approach typically relates different morphotypes of *E. huxleyi* to different degrees of calcification based on visual impression. *E. huxleyi* is a morphologically diverse species with several recognized morphotypes [15, 34, 35], potentially representing distinct genetic varieties [36]. Different *E. huxleyi* morphotypes display environmental preferences over both seasonal (e.g. [33, 37]) and regional (e.g. [8, 15, 32, 38–40]) gradients, and might furthermore differ in their sensitivity to ocean acidification [41]. *E. huxleyi* morphotypes differ in robustness of distal shield elements and degree of central area closure, which is often interpreted as differences in degrees of calcite production or differences in coccolith mass between the morphotypes. For example, [32] related the relative abundance of *E. huxleyi* Type B/C morphotypes to the degree of calcite production in the Southern Ocean, and reported based on the abundance of Type B/C that *E. huxleyi* calcification had not changed over 12 years in this region. Meanwhile, [33] reported an increase in *E. huxleyi* calcite production with decreasing pH in the Bay of Biscay. This was based on the increased relative abundance of "Overcalcified" Type A with decreasing pH, assuming that the "Overcalcified" morphotype is systematically heavier than the "normally calcified" Type A. Moreover, the dominance of Type A in North Atlantic *E. huxleyi* blooms and Type B/C in *E. huxleyi* blooms at the Patagonian Shelf have been suggested to explain differences in calcite production between these blooms [42], while [8] linked increasing *E. huxleyi* coccolith mass in the Chilean upwelling zone with an increasing relative abundance of "Overcalcified" *E. huxleyi* morphotypes.

In contrast to these studies, [26] found similar coccolith mass (relative to size) for Type A and Type B coccoliths, despite the more delicate appearance of Type B coccoliths. Moreover, a

recent laboratory study reported that only one out of two strains of "Overcalcified" *E. huxleyi* morphotypes had a higher calcite content per cell than other *E. huxleyi* strains [39], and [43, 44] found that high degrees of malformation did not affect cellular calcite content in *E. huxleyi*.

The usefulness of *E. huxleyi* morphotypes as proxies for coccolith mass or degree of calcite production in *E. huxleyi* is unclear and needs to be validated. The goal of this study is thus to investigate.

1. whether coccoliths of different *E. huxleyi* morphotypes differ systematically in thickness and mass,

2. whether changes in relative abundance of *E. huxleyi* morphotypes between samples can be used as a proxy for changes in mean *E. huxleyi* coccolith mass between samples,

3. whether relative abundance of *E. huxleyi* morphotypes in a sample can be used as a proxy for calcite production at the sampled site.

## Oceanographic setting

This study was conducted using samples collected near the Canary Islands, off the northwestern African coast during the two cruises F.S. METEOR 37/2b [45] and F.S. POSEIDON 233a [46] as part of the CANIGO project [47]. The Canary Islands are situated around 28-29˚N in the North Atlantic in the southwest-moving Canary Current in a transitional zone between the northwest African coastal upwelling region and the oligotrophic region of the North Atlantic subtropical gyre [48]. Samples were collected at two stations during both cruises (Figs 1 and 2): EBC, situated near the coast at 28˚42.5'N and 13˚09.3'W, and LP, situated further from the coast at 29˚45.7'N and 17˚57.3'W [49, 50]. The EBC station is thus located within an upwelling area, while LP is located in an oligotrophic area [51, 52]. [53] identified two distinct seasons with regards to the upwelling in this region, with April-September giving the most favourable upwelling conditions leading to maximal upwelling intensity in July. The other season from October to March gives less favourable upwelling conditions and a minimum in upwelling intensity in December and January. Sea surface temperature varies more in the upwelling region, ranging from as low as 17˚C in winter and early spring to as high as 24˚C in summer and fall, while temperatures generally range between 20 and 24˚C offshore near LP [51, 53]. Generally temperature differs between the coastal upwelling region and offshore by around 3-4˚C [51, 53], though the difference can decrease to less than 1˚C during winter [51]. Salinity has been reported to vary by up to ~0.5 units between coast and offshore [54]. During upwelling, filaments of upwelling waters can bring cold upwelling water and organic matter several hundred kilometers offshore [48, 51]. However, [51] found that while these filaments did at times reach LP as evidenced by reduced SST, primary productivity did not change at these times, indicating that the filament water may be nutrient depleted by the time it reaches LP.

Biological activity is very high in the coastal area near EBC, and chlorophyll $\alpha$ and primary productivity values are significantly higher at EBC compared to LP [50, 51]. [51] reported for example chlorophyll $\alpha$ values ranging from 0.03-0.25mgm$^{-3}$ at LP between September 1997 and March 1999, while chlorophyll $\alpha$ values at EBC over the same time period ranged from ~0.07-0.89mgm$^{-3}$. Meanwhile, primary productivity ranged from 0.2-0.7gCm$^{-2}$d$^{-1}$ at LP over the September 1997 to March 1999 time period, while it ranged from 0.4-1.2gCm$^{-2}$d$^{-1}$ at EBC [51]. Primary production is typically high during late winter and early spring both at EBC and LP, due to a yearly phytoplankton bloom during this time period [51]. At EBC upwelling also leads to additional primary productivity peaks throughout the year [51].

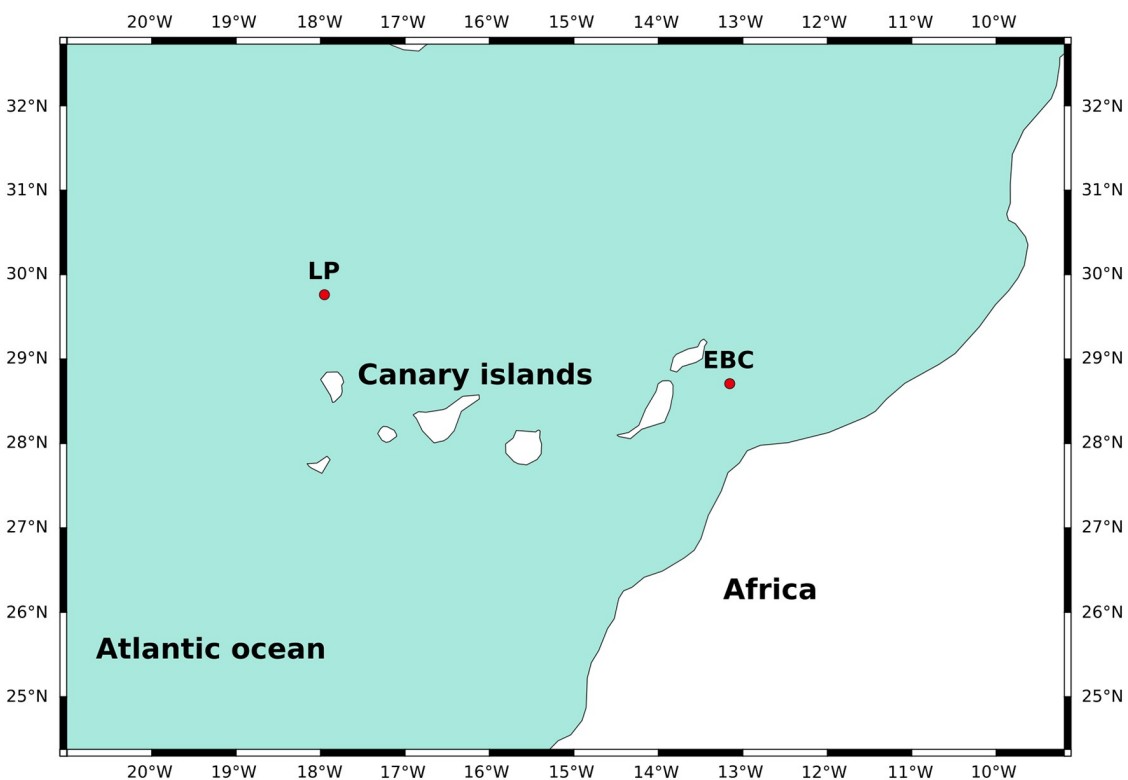

**Fig 1. Location of sample sites LP1 and EBC2 (red circles) in the Canary Islands region off the coast of Northwest Africa.** Map was made with Natural Earth.

## Materials and methods

Sea water was collected in ten litre Niskin bottles and filtered onto 47mm diameter Nucleopore filters with 0.8 μm diameter pores. The filters were rinsed after filtration with $NH_4OH$-buffered distilled water with a pH of 8.5 (see [56] for details). Sample depths for this study (15m, 50m, and 100m at the open ocean site in January, 10m at the coastal site in January, 10m at the open ocean site in September, and 75m at the coastal site in September) were chosen based on *E. huxleyi* cell densities from [55] (Fig 3). Specifically, the samples with the highest cell density at each site was chosen for analysis, plus samples at 15m and 50m at the open ocean site in January to evaluate potential changes with depth. The top most sample from each location was chosen for further MCMC analysis (see below for details).

### SEM analysis

A triangular piece was cut from each filter membrane, mounted on a stub and coated with platinum using a Leica SCD500 Metal Coater (Leica Microsystems, Wetzlar, Germany). Up to 1500 images per sample were then automatically captured at 3000x magnification with 1024 x 768 pixel resolution using a Zeiss Supra VP55 SEM (Carl Zeiss, Oberkochen, Germany). *E. huxleyi* coccoliths were counted and classified into various morphotypes and/or morphological groups. Type A and "Overclassified" Type A were classified according to [57] and [58]. Coccoliths belonging to Type C or Type B/C were grouped together and counted as Group B according to [58]. Malformed coccoliths, where distal shields were irregularly formed, often with disconnected distal shield elements and/or a missing/minimal central tube, were counted

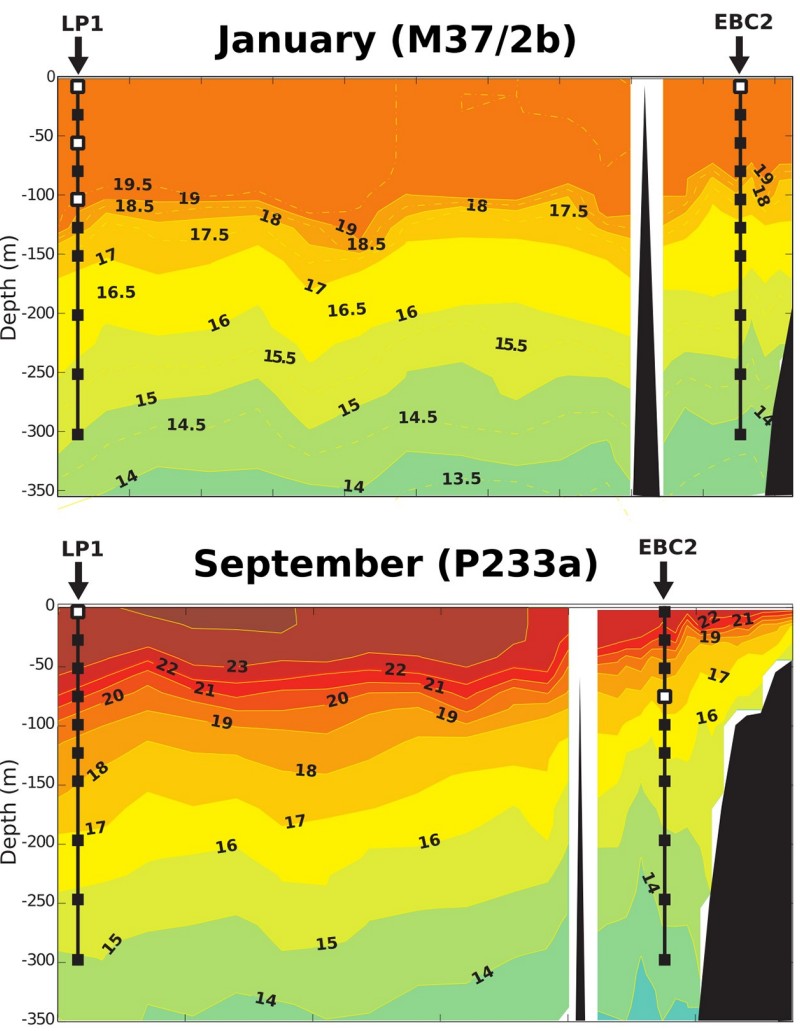

**Fig 2. Temperature profile in January and September 1997 (in ˚C) along the 29˚latitude transect.** Black squares: sampled depths during cruises M37/2b and P233a; Black squares with white inlet: sampled depths investigated in this study. Figure modified from [55].

as malformed (Fig 4). The lengthand width of all counted coccoliths were measured using ImageJ 1.52a after calibration of the SEM using spherical beads with a 2 μm ±0.1 μm diameter (DYNO Particles, Lillestrøm, Norway). Broken or partially covered coccoliths were counted and measured if their length could be accurately determined.

The length and width were measured for each counted coccolith in the shallowest samples at each site (Samples: M37/2b-31-10m—Coastal January, M37/2b-50-15m—Open ocean January, P233a-600-75m—Coastal September, and P233a-582-10m—Open ocean September). The central tube width along the length axis for up to 30 Type A coccoliths in each sample was measured as well (Fig 5). The central tube width was measured along the length axis (similar to [30]). To account for variation in central tube width, central tube width was measured in the present study on both sides of the central area, and then averaged to give a mean central tube width. The central tube width was then divided by coccolith length to obtain a size-independent central tube width to length ratio (CT:L).

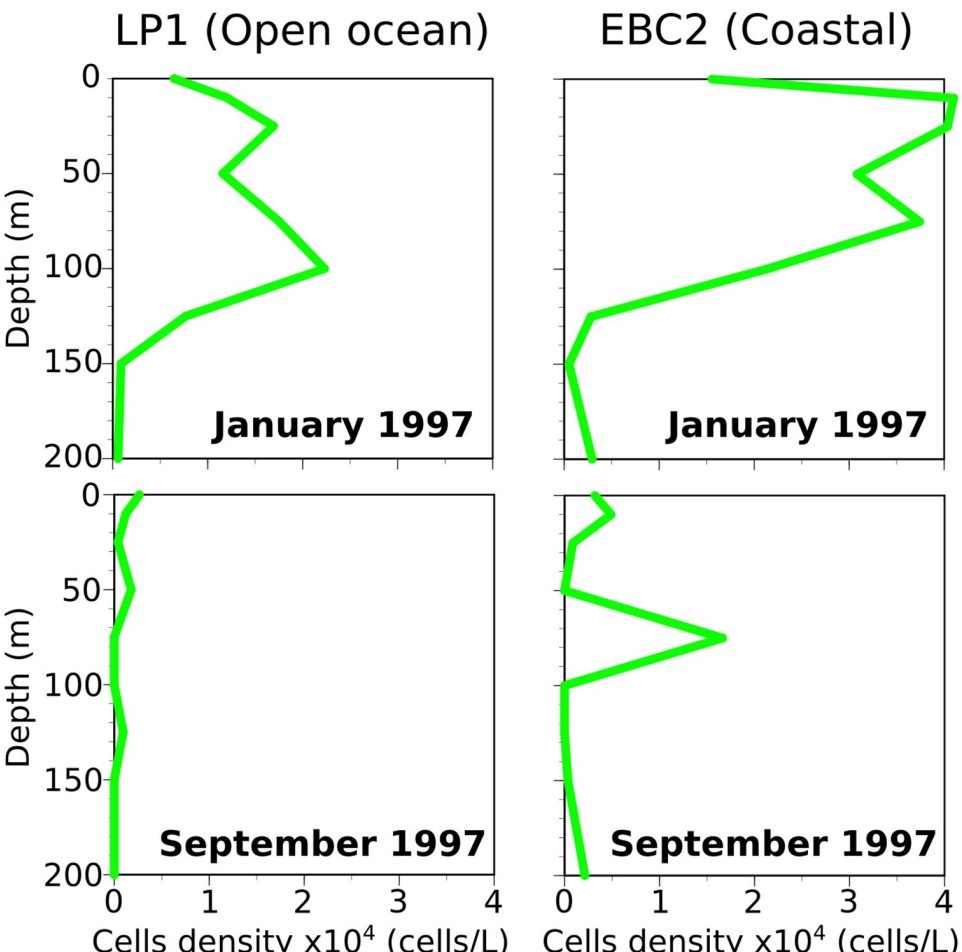

**Fig 3. *E. huxleyi* cell density at each station measured during cruises M37/2b (January 1997) and P233a (September 1997).** Data from [55].

## Coccolith mass and thickness analysis

Coccoliths were transferred from filters onto glass slides and mounted using NOA 61 adhesive. Coccolith images were then captured using a Zeiss Axio Imager Z1 light microscope (Zeiss, Oberkochen, Germany) equipped with a Benford plate for circular polarization [59], a 1.6x optovar, neutral density filters, a Plan-Apo 100x, 1.4 NA oil objective, a 0.9 NA universal condenser, and a Canon 60D DSLR camera (Canon Inc., Tokyo, Japan) with a 5194 x 3457 resolution (a pixel resolution of 0.0003 $\mu m^{-2}$) for light microscope (LM) analysis. Images were captured in RAW and converted to TIFF in sRGB colour space with gamma 2.2 according to [60]. Size calibration was done using a S8 Stage micrometer (02A00404 from PYSER-SGI Ltd., Edenbridge, UK) with steps 10 μm apart along a line with an overall length of 1000 μm ±1 μm. Microscope illumination and camera sensitivity were calibrated for accurate retardation / thickness estimation using polymer films with known retardations of 31nm and 129nm according to the Circular Polarizer Retardation estimates (CPR) method [61]. The condenser was partly closed to avoid polarization aberrations [62, 63]. Grey values were related to an sRGB Michel-Lévy chart from [60] using the ImageJ function [Calibrate…] in ImageJ 1.52. Subsequently, coccolith length, mass, and thickness were measured in ImageJ according to the

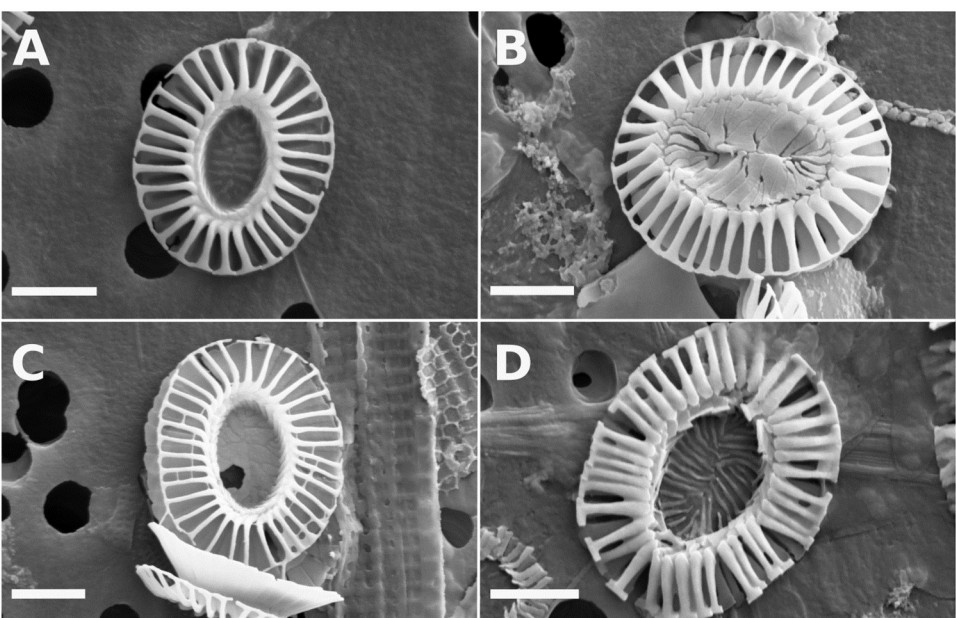

**Fig 4. *E. huxleyi* morphotypes.** A: Type A. B: "Overcalcified" Type A. C: Group B. D: malformed coccoliths. Horizontal bar: scale = 1 µm.

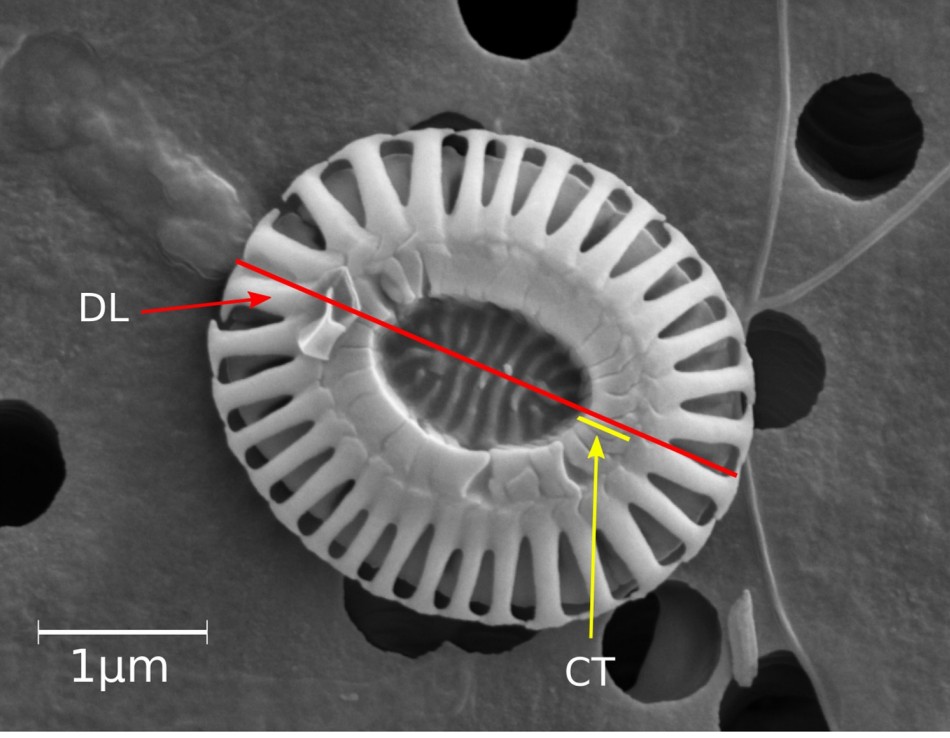

**Fig 5. SEM image of *E. huxleyi* Type A coccolith.** Red line shows the axis of distal shield (DL), the yellow line shows central tube width (CT). CT was measured at both sides of the central area along the DL axis and averaged.

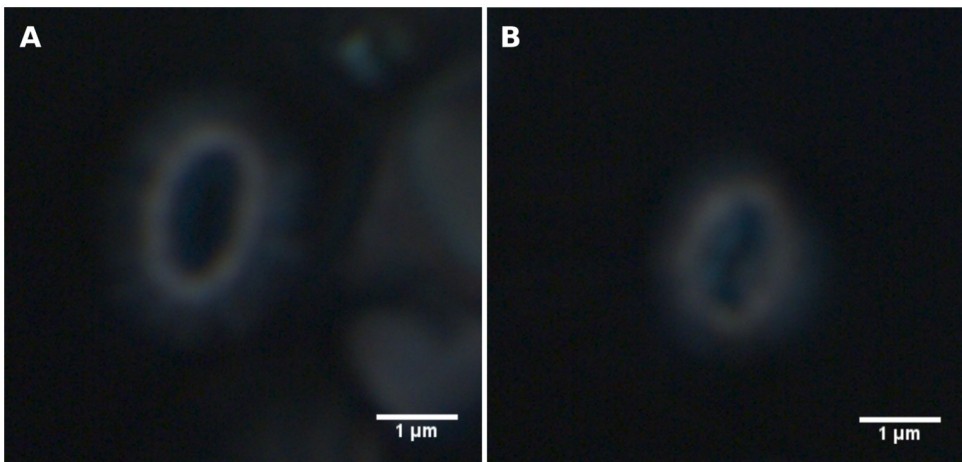

**Fig 6. *E. huxleyi* coccoliths captured under circular polarized light.** Note the different appearance of their central areas. A: *E. huxleyi* coccolith with a normal central area. B: *E. huxleyi* coccolith with an overgrown central area.

CPR-method [61]. Coccoliths were separated from the background using a Canny-Deriche edge detection algorithm [64, 65] with an $\alpha$ of 0.5, before the ImageJ function [Analyze particles. . .] was used to obtain coccolith length, mass, and thickness. The standard uncertainty at a 95% confidence level of the LM measurements is ±0.2 μm for length, ±0.007 μm for thickness, and ±~14-15%/~0.3-0.4pg (depending on size) for mass.

"Overcalcified" Type A coccoliths could be recognized in the light microscope due to its closed central area (Fig 6, see also [66]). "Overcalcified" Type A coccolith thickness and mass could therefore be measured directly. However, Type A, Group B, and malformed coccoliths could not be distinginguished in the light microscope, and the mean thickness of these morphotypes had to be estimated from a Markov Chain Monte Carlo (MCMC) analysis (see below for details). Sample sizes were adjusted to obtain approximately 30 coccoliths of the rarest morphotype (see Table 1). Please note that only 75 coccoliths were measured from sample P233a-582-10m (Open ocean September) due to its coccolith scarcity.

## Statistical analysis

All statistical analyses in this study were done using R version 3.4.3 [67] in RStudio version 1.1.383. Mann-Whitney U-tests were performed to evaulate differences in coccolith length, mass, and thickness between months and stations while a Kruskal-Wallis test was performed to evaluate differences in coccolith length, mass, and thickness with depth. t-tests were performed to evaluate differences in length and central tube width of morphotypes between stations and seasons.

**Markov chain monte carlo analysis.** A Markov Chain Monte Carlo (MCMC) model was used to estimate mean thickness values for different *E. huxleyi* morphotypes at each site. Assuming that the thickness of each individual *E. huxleyi* morphotype population is normally distributed in each sample, the measured *E. huxleyi* coccolith thickness represents a mixed distribution of a number of normal distributions equal to the number of distinct morphotypes. The contribution (mixture weight) of each individual normal distribution in the mixed distribution would then correspond to the relative abundance of the individual morphotypes. The task of the MCMC analysis was therefore to estimate the mean and standard deviation values for the individual normal distributions using the relative abundance (mixture weight) and the total (mixed) distribution as input.

**Table 1. Coccolith morphotype counts, relative abundance, length measurements, and central tube: Length ratio (CT:L) for the most shallow sample at each site measured in this study.** All measurements and counts were obtained from SEM images. Mean length and mean CT:L are shown including the 95% confidence interval (CI 95%). N: Number of counted coccoliths. Rel. ab. (%): Relative abundance of morphotypes. Morphotype abbreviations: **A**: Type A, **OA**: Overcalcified Type A, **B**: Group B, **M**: Malformed coccoliths.

| Sample | Morphotype | N | Rel. ab. (%) | Length (µm) Mean | Length (µm) CI 95% | Width (µm) Mean | Width (µm) CI 95% | Aspect ratio Mean | Aspect ratio CI 95% | CT:L Mean | CT:L CI 95% |
|---|---|---|---|---|---|---|---|---|---|---|---|
| M37/2b-31-10m | A | 231 | 77 | 3.36 | 0.04 | 2.79 | 0.03 | 1.21 | 0.01 | 0.07 | 0.01 |
| | B | 67 | 22 | 3.18 | 0.09 | 2.66 | 0.09 | 1.20 | 0.02 | – | – |
| | M | 2 | 1 | 2.77 | – | 2.23 | – | 1.24 | – | – | – |
| | Total | 300 | 100 | 3.32 | 2.75 | 0.03 | 0.04 | 1.21 | – | – | – |
| M37/2b-50-15m | A | 264 | 88 | 3.41 | 0.04 | 2.84 | 0.04 | 1.20 | 0.01 | 0.07 | 0.01 |
| | B | 36 | 12 | 3.16 | 0.15 | 2.63 | 0.12 | 1.20 | 0.01 | – | – |
| | Total | 300 | 100 | 3.38 | 0.04 | 2.81 | 0.04 | 1.20 | 0.00 | – | – |
| P233a-600-75m | A | 108 | 36 | 3.14 | 0.06 | 2.61 | 0.06 | 1.21 | 0.01 | 0.06 | 0.01 |
| | OA | 56 | 19 | 3.00 | 0.08 | 2.51 | 0.06 | 1.21 | 0.01 | – | – |
| | B | 114 | 38 | 2.95 | 0.07 | 2.44 | 0.06 | 1.20 | 0.01 | – | – |
| | M | 22 | 7 | 2.98 | 0.14 | 2.39 | 0.13 | 1.25 | 0.04 | – | – |
| | Total | 300 | 100 | 3.03 | 0.04 | 2.51 | 0.04 | 1.21 | 0.01 | – | – |
| P233a-582-10m | A | 24 | 8 | 3.23 | 0.12 | 2.62 | 0.11 | 1.25 | 002 | 0.04 | 0.00 |
| | B | 2 | 1 | 2.94 | – | 2.50 | – | 1.17 | – | – | – |
| | M | 262 | 91 | 3.20 | 0.04 | 2.59 | 0.03 | 1.24 | 0.01 | – | – |
| | Total | 288 | 100 | 3.21 | 2.59 | 0.03 | 0.04 | 1.24 | 0.01 | – | – |

In other words, mean thickness for each morphotype was estimated by fitting the frequency distribution of thickness measurements to the relative abundance of *E. huxleyi* morphotypes. This was done using a Bayesian mixed normal distributions model built with a Gibbs sampler MCMC algorithm [68] using JAGS version 4.3.0 in R with the package R2Jags [69] and a template written by [70]. The template represent a model with *K* number of components (i.e. morphotypes), where each component is a normal distribution with a sampled mean and standard deviation. The model is initiated with prior distributions specified for each of the parameters mean, precision (defined as $\frac{1}{variance}$ [71]), and relative abundance. Standard deviation is then calculated from the estimated precision of the model. The template of [70] was modified as follows:

1. The model samples for precision rather than standard deviation to keep with normal practice for the model software [71]. Standard deviation is then calculated from the precision as

$$Standard\ deviation = \sqrt{\frac{1}{precision}} \tag{1}$$

2. The precision of the prior distribution of the mean parameter was altered to sample the mean parameter more efficiently.

3. Standard deviation/precision is sampled for each individual component, rather than sampling one constant standard deviation value for all components.

4. A gamma distribution is used instead of a t-distribution for the precision prior distribution.

5. Probability of component assignment for each data item is determined from a gamma distribution with the shape parameter $\alpha$ determined according to a dirichlet prior distribution

and scale parameter $\theta$ set to 1 (see [71] for details). To avoid infinity issues during sampling of the four components model a minimum limit of 0.01 was set for the gamma values. Alpha hyperpriors for the dirichlet prior distribution are the prior relative abundance parameters, and were set as 20 and 80 for the two components models and 10, 35, 35, and 20 for the four components model.

6. No sorting of the mean parameter (two-component models only).

Outliers, defined as coccoliths with thickness >1.5 times the interquartile range, were removed from each sample before running the MCMC, and morphotypes with relative abundances <1% in a given sample were ignored for the model. The model for each sample was run for 500,000 iterations with a 50,000 burn-in period and a thinning interval of 100. Convergence was confirmed visually using a trace plot (S1, S2, S3 and S4 Figs) and formally using the Gelman-Rubin diagnostic. Estimated mean and standard deviation of each morphotype was determined from the median of each sampled parameter with a credible interval showing the region around the median which 68% of the parameter samples fell within (a 68% credible interval is roughly analogous to a 1 standard deviation margin of error/confidence interval [72]).

The accuracy and efficiency of the models were evaluated on eight groups of simulated samples (Figs 7 and 8). The samples were created according to different scenarios (Table 2) using a random number generator. Three simulated samples were generated for each two-component scenario, while one simulated sample was generated for each four-component scenario. This process allowed for the model to be calibrated and evaluated under various conditions where the actual mean and standard deviation values for each component are known.

## Morphotype mass estimation

Assuming an elliptical coccolith shape, the mean coccolith mass of different coccolith morphotypes was approximated according to this formula:

$$m = (\pi \times \frac{l}{2} \times \frac{w}{2}) \times t \times d \tag{2}$$

where $m$ is coccolith mass, $l$ is coccolith length, $w$ is coccolith width, $t$ is coccolith thickness, and $d$ is calcite density (= 2.71g cm$^{-3}$).

## Calcite concentration estimation

Total *E. huxleyi* calcite concentration at each site was estimated from *E. huxleyi* cell density and coccolith mass according to the following equation:

$$Ca_{EHUX} = CD \times m \times N_{cocco.} \tag{3}$$

where $Ca_{EHUX}$ is *E. huxleyi* calcite concentration in μgL$^{-1}$, $CD$ is cell density, $m$ is mean coccolith mass, and $N_{cocco.}$ is the number of coccoliths per cell. Cell density data was obtained from [55], while $N_{cocco.}$ was taken from [73] and [74]. Three different coccoliths per coccosphere numbers were used to give a probable range: 23 (mean number in both [73] and [74]), 10 (minimum in [74]), and 48 (maximum in [74]). The number of coccoliths per coccosphere was for simplicity assumed to be the same for all morphotypes.

## Results

Overall *E. huxleyi* coccolith length, thickness, and mass was determined using light microscope (LM) images of a total of 1091 *E. huxleyi* coccoliths (Fig 9, Table 3). Mean coccolith length

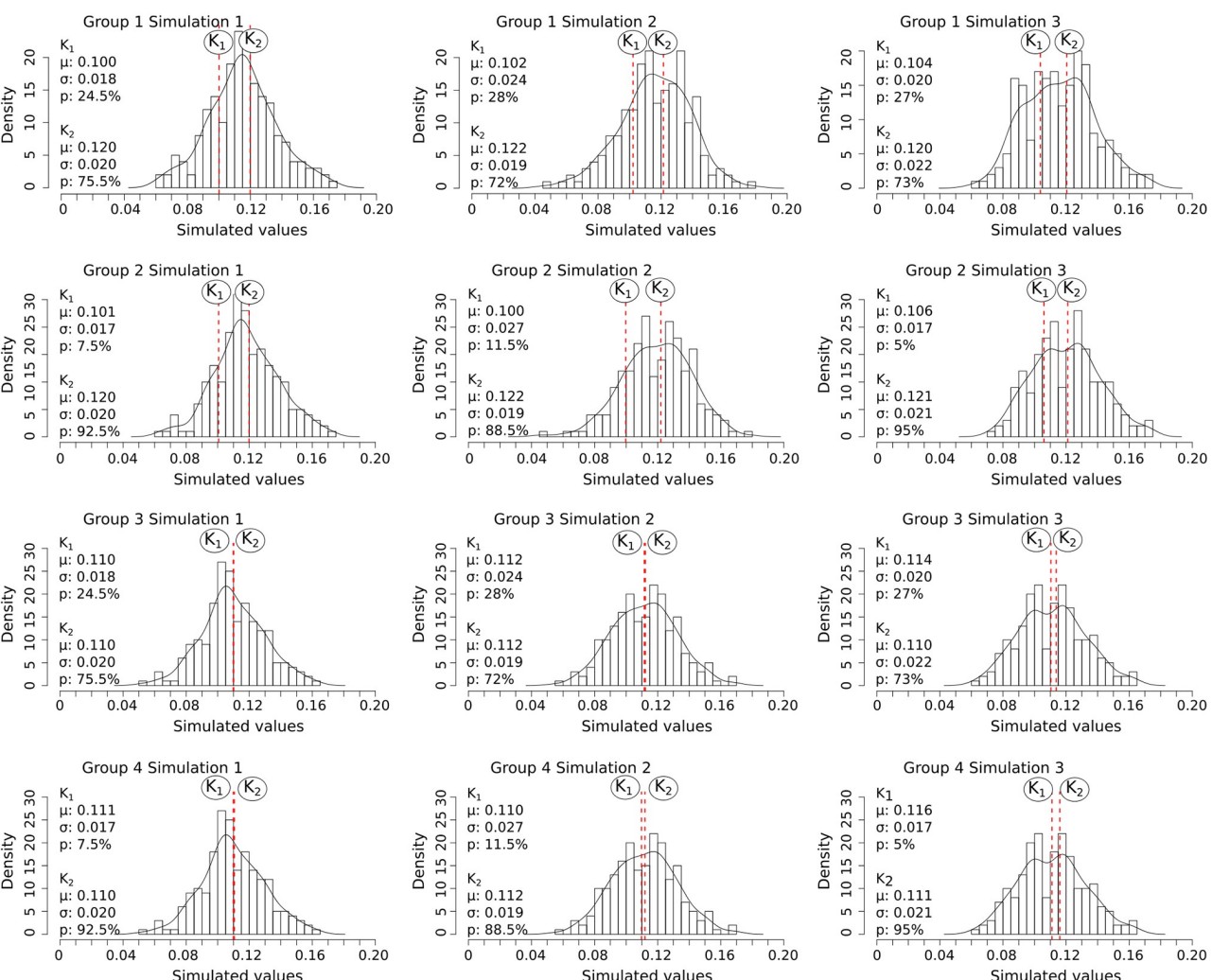

**Fig 7. Density bar plots showing the distribution of simulated values for the two-component samples.** Black curve: density curve for each sample; vertical red stapled lines: the position of the mean of each component (simulated morphotype) $K_n$. $K_1$: component/morphotype 1. $K_2$: component/ morphotype 2. $\mu$: the component mean. $\sigma$: the component standard deviation. $p$: the relative abundance.

from LM images ranged from a maximum of 3.4 μm at the open ocean site in January to a minimum of 3.1 μm at the coastal site in September (Fig 9, Table 3). These length values compare well with the corresponding sample coccolith lengths measured from SEM images (Table 1). Coccolith thickness differed significantly from a maximum of 0.150 μm at the open ocean site in January to a minimum of 0.093 μm at the open ocean site in September (Fig 9, Tables 3 and 4). Mean *E. huxleyi* coccolith mass varied from a maximum of 2.6pg at the open ocean station in January to a minimum of 1.7pg at both stations in September.

Total calcite concentration of the *E. huxleyi* population was estimated assuming 10, 23, or 48 coccoliths per cell (Fig 10). At 23 coccoliths per cell, total *E. huxleyi* calcite ranged from 0.05 to $2.26\mu gL^{-1}$, while with 10 coccoliths per cell it ranged from 0.02 to $0.98\mu gL^{-1}$ and with 48 coccoliths per cell the range was 0.10 to $4.72\mu gL^{-1}$ (Table 5). *E. huxleyi* calcite concentration was higher in January than September at both sites. At the coastal site total *E. huxleyi* calcite concentration decreased from $2.26\mu gL^{-1}$ to $0.65\mu gL^{-1}$, while at the open ocean site it decreased from $0.74\mu gL^{-1}$ to $0.05\mu gL^{-1}$.

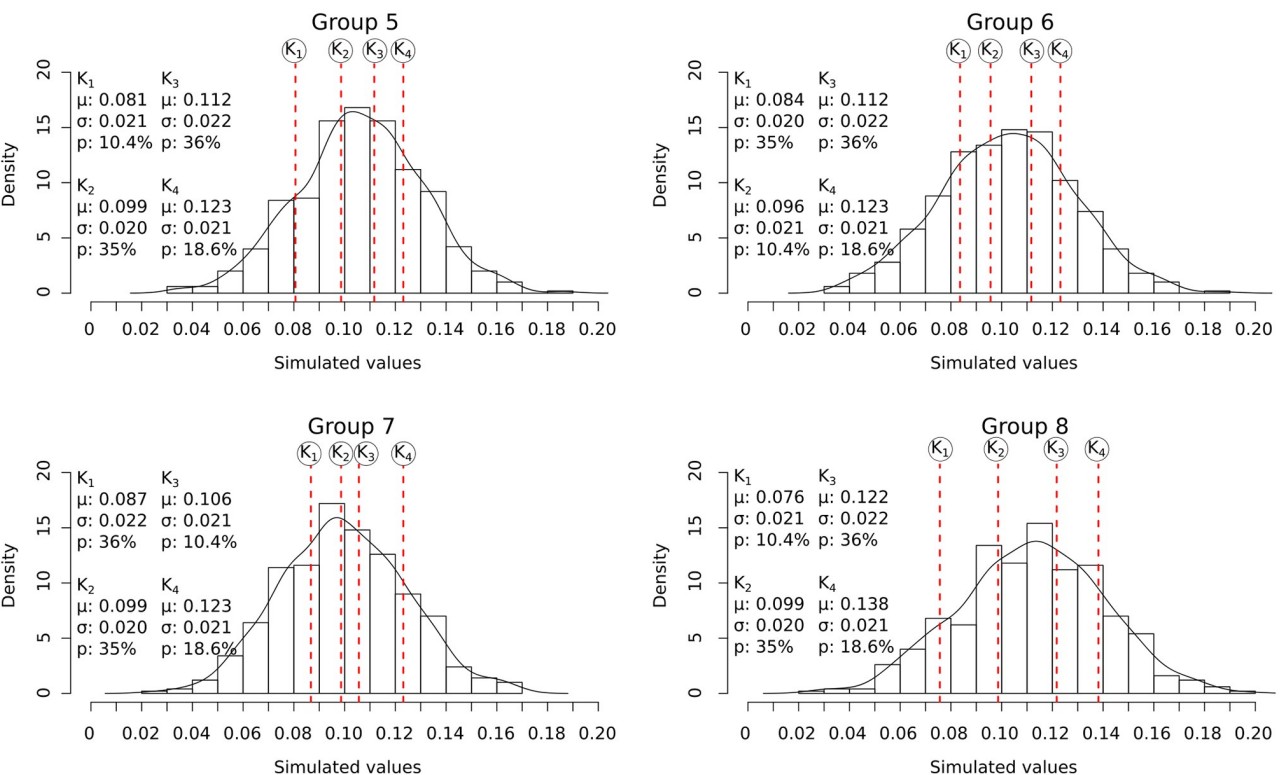

**Fig 8. Density bar plots showing the distribution of simulated values for the four-component samples.** Black curve: the density curve for each sample, vertical red stapled lines: the position of the mean of each component (simulated morphotype) $K_n$. $K_1$: component/morphotype 1. $K_2$: component/morphotype 2. $K_3$: component/morphotype 3. $K_4$: component/morphotype 4. $\mu$: the component mean. $\sigma$: the component standard deviation. $p$: the relative abundance.

Coccolith length, thickness, and mass obtained from LM were also evaluated along a depth transect from 10 to 100 meters depth at the open ocean station (LP) in January. Thickness and mass varied statistically significantly with depth (Kruskal-Wallis $p < 0.05$, Tables 3 and 6), though only thickness varied by a degree larger than the standard uncertainty of measurements (see Methods section). Thickness was greatest at 50m depth (0.149 μm), while thickness was similar at 15m and 100m depth (0.133 μm and 0.126 μm, respectively).

**Table 2. Conditions for each simulated sample as assigned to a random number generator.** 1 to 3 samples were generated according to conditions in each row. Sim.: Number of simulated samples generated in each group; N: sample size; k: number of components (morphotypes); $RA_n$: Relative abundance of each component n; $T_n$: Mean thickness of each simulated component n; SD: standard deviation of the sample distribution.

| Group | Sim. | N | k | RA₁ (%) | RA₂ (%) | RA₃ (%) | RA₄ (%) | T₁ | T₂ | T₃ | T₄ | SD |
|-------|------|-----|---|---------|---------|---------|---------|-------|-------|-------|-------|-------|
| 1 | 3 | 200 | 2 | 25 | 75 | – | – | 0.100 | 0.120 | – | – | 0.020 |
| 2 | 3 | 200 | 2 | 10 | 90 | – | – | 0.100 | 0.120 | – | – | 0.020 |
| 3 | 3 | 200 | 2 | 25 | 75 | – | – | 0.110 | 0.110 | – | – | 0.020 |
| 4 | 3 | 200 | 2 | 10 | 90 | – | – | 0.110 | 0.110 | – | – | 0.020 |
| 5 | 1 | 500 | 4 | 10 | 35 | 35 | 20 | 0.085 | 0.100 | 0.110 | 0.125 | 0.02 |
| 6 | 1 | 500 | 4 | 35 | 10 | 35 | 20 | 0.085 | 0.100 | 0.110 | 0.125 | 0.02 |
| 7 | 1 | 500 | 4 | 35 | 35 | 10 | 20 | 0.085 | 0.100 | 0.110 | 0.125 | 0.02 |
| 8 | 1 | 500 | 4 | 10 | 35 | 35 | 20 | 0.080 | 0.100 | 0.120 | 0.140 | 0.02 |

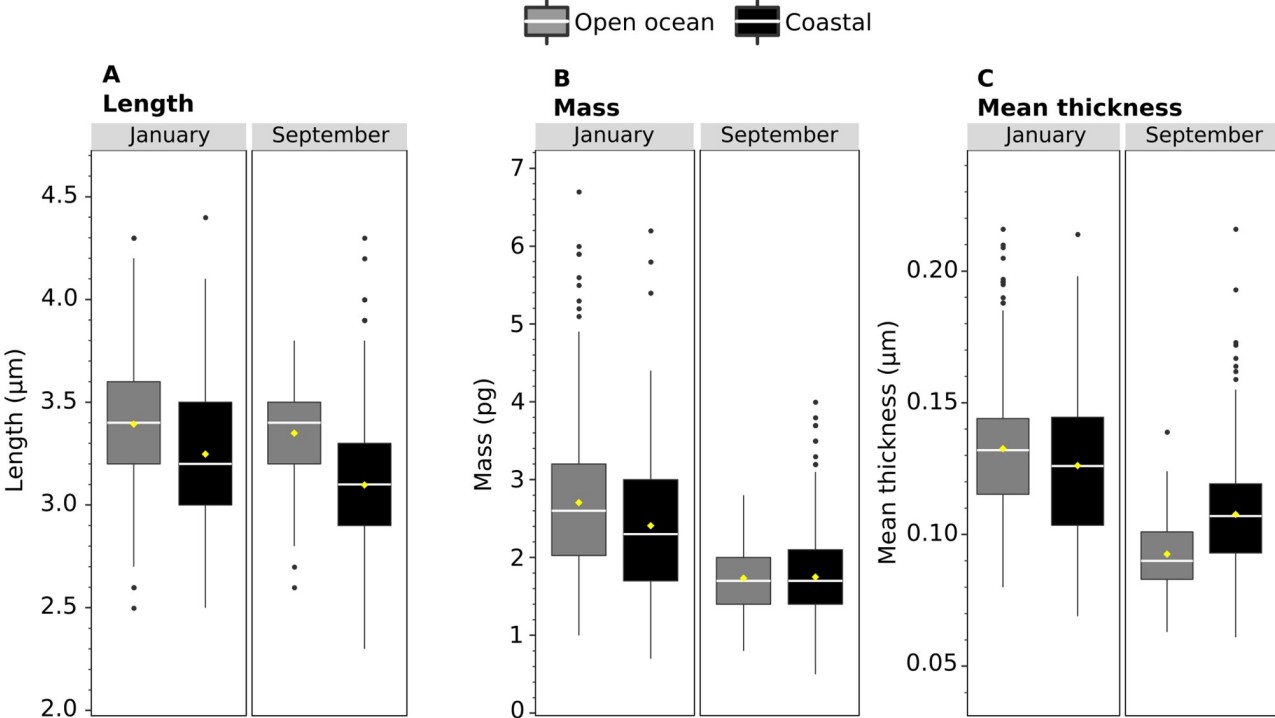

**Fig 9. Boxplots showing length, mass, and thickness of *E. huxleyi* coccoliths measured on the light microscope.** January represents samples from 10m at EBC (coastal) and 15m at LP (offshore) during METEOR cruise 37/2b, while September measurements represent samples from 75m at EBC (coastal) and 10m at LP (offshore). A: length. B: mass. C: mean thickness per coccolith. Horizontal white lines indicate the median, while yellow diamonds are the average values for each sample. Black dots represent outliers (values greater than 1.5 times the interquartile range).

## Individual *E. huxleyi* morphotypes measurements

In order to analyze the relationship between *E. huxleyi* morphotypes and coccolith mass, the relative abundance of *E. huxleyi* morphotypes at each site was determined from SEM images (Fig 11). At least four different morphotypes were identified from a total of 1188 *E. huxleyi* coccoliths: Type A, "Overcalcified" Type A, Group B, and malformed coccoliths (Fig 4, Table 1). January samples consisted of Type A and Group B coccoliths, while September samples also contained significant numbers of malformed coccoliths at both sites, and "Over-calcified" Type A at the coastal site. Type A dominated both sites in January (relative

**Table 3. Mean values for length, mass, and mean thickness obtained on a light microscope.** N: Sample size; CI 95%: 95% confidence interval for population mean. Numbers in italics represent subsets of either "open" central area *E. huxleyi* coccoliths or "closed" central area *E. huxleyi* in the sample were both were present.

| Sample | N | Length (µm) | | Mass (pg) | | Mean thickness (µm) | |
|---|---|---|---|---|---|---|---|
| | | Mean | CI 95% | Mean | CI 95% | Mean | CI 95% |
| M37/2b-31-10m | 143 | 3.2 | 0.1 | 2.4 | 0.2 | 0.126 | 0.005 |
| M37/2b-50-15m | 250 | 3.4 | 0.1 | 2.7 | 0.1 | 0.133 | 0.003 |
| M37/2b-50-50m | 61 | 3.3 | 0.1 | 2.9 | 0.3 | 0.150 | 0.007 |
| M37/2b-50-100m | 61 | 3.3 | 0.0 | 2.5 | 0.3 | 0.126 | 0.008 |
| P233a-600-75m | 424 | 3.1 | 0.0 | 1.7 | 0.1 | 0.108 | 0.002 |
| P233a-600-75m "open" | *359* | *3.1* | *0.0* | *1.7* | *0.1* | *0.104* | *0.002* |
| P233a-600-75m "Overcalcified" | *70* | *3.1* | *0.1* | *2.0* | *0.2* | *0.124* | *0.006* |
| P233a-582-10m | 75 | 3.3 | 0.1 | 1.7 | 0.1 | 0.093 | 0.003 |

**Table 4. Mann-Whitney U-test comparing length, central tube width to Coccolith Length ratio (CT:L—Type A only), mass and thickness at the shallowest samples at each site.** Italicized font indicates *p*-values that could not be computed exactly due to ties and were instead computed according to a normal approximation.

| Length | | | | |
|---|---|---|---|---|
| Samples compared | $N_1$ | $N_2$ | U | *p* |
| M37/2b-31-10m (Coastal January) and P233a-600-75m (Coastal September) | 300 | 300 | 65256 | <0.01 |
| M37/2b-50-15m (Open ocean January) and P233a-582-10m (Open ocean September) | 300 | 288 | 55946 | <0.01 |
| M37/2b-31-10m (Coastal January) and M37/2b-50-15m (Open ocean January) | 300 | 300 | 39996 | 0.02 |
| P233a-600-75m (Coastal September) and P233a-582-10m (Open ocean September) | 300 | 288 | 30265 | <0.01 |
| CT:L | | | | |
| Samples compared | $N_1$ | $N_2$ | U | *p* |
| M37/2b-31-10m (Coastal January) and P233a-600-75m (Coastal September) | 30 | 30 | 600 | *0.03* |
| M37/2b-50-15m (Open ocean January) and P233a-582-10m (Open ocean September) | 30 | 24 | 27 | <0.01 |
| M37/2b-31-10m (Coastal January) and M37/2b-50-15m (Open ocean January) | 30 | 30 | 419 | 0.65 |
| P233a-600-75m (Coastal September) and P233a-582-10m (Open ocean September) | 30 | 24 | 9 | <0.01 |
| Mass | | | | |
| Samples compared | $N_1$ | $N_2$ | U | *p* |
| M37/2b-31-10m (Coastal January) and P233a-600-75m (Coastal September) | 143 | 424 | 43242 | <0.01 |
| M37/2b-50-15m (Open ocean January) and P233a-582-10m (Open ocean September) | 250 | 75 | 15661 | <0.01 |
| M37/2b-31-10m (Coastal January) and M37/2b-50-15m (Open ocean January) | 143 | 250 | 14898 | 0.01 |
| P233a-600-75m (Coastal September) and P233a-582-10m (Open ocean September) | 424 | 75 | 15537 | 0.75 |
| Mean thickness | | | | |
| Samples compared | $N_1$ | $N_2$ | U | *p* |
| M37/2b-31-10m (Coastal January) and P233a-600-75m (Coastal September) | 143 | 424 | 41909 | <0.01 |
| M37/2b-50-15m (Open ocean January) and P233a-582-10m (Open ocean September) | 250 | 75 | 17426 | <0.01 |
| M37/2b-31-10m (Coastal January) and M37/2b-50-15m (Open ocean January) | 143 | 250 | 15452 | 0.03 |
| P233a-600-75m (Coastal September) and P233a-582-10m (Open ocean September) | 424 | 75 | 23192 | <0.01 |

abundance >50%), but not in September. The open ocean site was dominated by malformed coccoliths in September, while the coastal site was not dominated by any single morphotype. The most common morphotype in September at the coastal site was Group B with a relative abundance of 38%, followed by Type A (36%), "Overcalcified" Type A (19%), and malformed coccoliths (7%) (Table 1).

Morphometric analysis of the same 1188 coccoliths on SEM images revealed that overall *E. huxleyi* coccolith length differed statistically significantly (Mann-Whitney *p* <0.05) at both

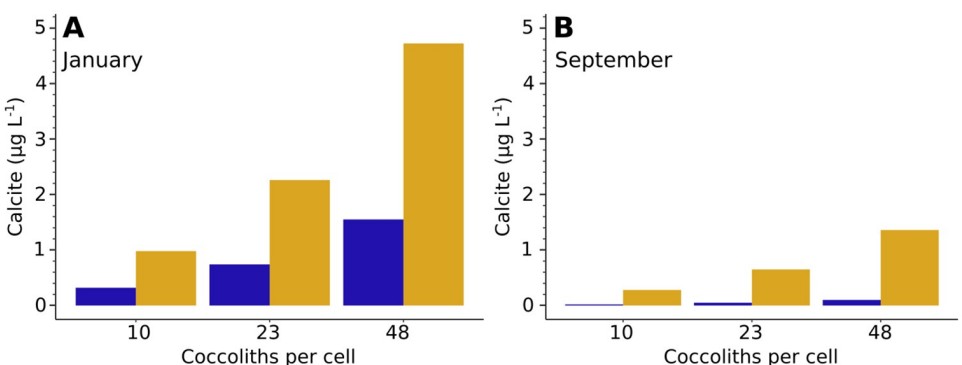

**Fig 10. Estimated *E. huxleyi* calcite concentration.** A: January. B: September. Blue: Open ocean site at 10 or 15m depth. Orange: Coastal site at 10 or 75m depth. Numbers of coccoliths per cell were taken from [73] and [74].

**Table 5. Estimated *E. huxleyi* calcite concentration at each site.** CD: Cell density; mass: Mean coccolith mass; TC$_{23}$: Total *E. huxleyi* calcite concentration assuming 23 coccoliths per cell [73, 74]; TC$_{10}$: Total *E. huxleyi* calcite concentration assuming 10 coccoliths per cell [74]; TC$_{48}$: Total *E. huxleyi* calcite concentration assuming 48 coccoliths per cell [74].

| Station | Month | Depth (m) | CD (cell/L) | mass (pg) | TC$_{23}$ (µgL$^{-1}$) | TC$_{10}$ (µgL$^{-1}$) | TC$_{48}$ (µgL$^{-1}$) |
|---------|-------|-----------|-------------|-----------|------------------|------------------|------------------|
| Coastal | January | 10 | 41003 | 2.4 | 2.26 | 0.98 | 4.72 |
| Open ocean | January | 15 | 11983 | 2.7 | 0.74 | 0.32 | 1.55 |
| Coastal | September | 75 | 16632 | 1.7 | 0.65 | 0.28 | 1.36 |
| Open ocean | September | 10 | 1228 | 1.7 | 0.05 | 0.02 | 0.10 |

sites between January and September (Table 4). *E. huxleyi* coccoliths were 0.29 μm longer in January at the coastal site and 0.17 μm longer in January at the open ocean site. Length furthermore differed significantly between the two sites both in January and September. Coccoliths at the open ocean site were 0.06 μm longer than coccoliths at the coastal site in January and 0.18 μm longer in September.

Type A coccolith length ranged from 3.14 μm to 3.41 μm (Table 1). Type A coccoliths were significantly longer in January than September at the coastal site (difference of 0.22 μm, t-test $p < 0.05$). However, there were no statistically significant differences between the two sites in either January or September. "Overcalcified" Type A coccoliths were only found at the coastal site in September where they were on average 3.00 μm long.

Group B coccolith length ranged from 2.95 μm to 3.18 μm (Table 1) and they were significantly longer (0.23 μm) in January than in September at the coastal site. Malformed coccoliths were significantly larger at the open ocean site than at the coastal site, with a coccolith length difference of 0.22 μm.

An MCMC analysis of thickness measurements of *E. huxleyi* coccoliths was used to estimate the thickness of individual morphotypes, as Type A, Group B, and malformed coccoliths could not be visually identified in the light microscope for direct thickness/mass measurements (Fig 6). Type A coccolith thickness ranged from 0.089 μm to 0.131 μm (Table 7). Type A coccoliths were thicker in January than September at both sites, with a thickness difference of 0.033 μm or 0.020 μm at the coastal site and 0.042 μm at the open ocean site.

Group B coccolith thickness ranged from 0.097 μm to 0.120 μm (Table 7) and coccoliths were 0.023 μm thicker at the open ocean site than at the coastal site in January. Meanwhile, coastal Group B coccolith thickness were similar in January and September (Fig 12). Malformed coccoliths were 0.091 μm thick at the open ocean site and 0.086 μm thick at the coastal site. While malformed coccoliths were thinner than other morphotypes at the coastal site, Type A and malformed coccoliths were similar in thickness at the open ocean site.

Because "Overcalcified" Type A coccolith mass could be determined directly from LM images (Fig 6), direct measurements were done of "Overcalcified" Type A at the coastal site in September. These measurements could be compared to LM measurements of the remaining sample, representing an aggregate of all other *E. huxleyi* morphotypes present in the sample (Table 3). They could also be compared to coccolith thickness and mass of other morphotypes

**Table 6. Kruskal-Wallis test results comparing length, mass and thickness at the depths 15m, 50m, and 100m at the open ocean station LP in January.** Measurements for all parameters were obtained from the light microscope images. df = degrees of freedom.

| Parameter | N$_{15m}$ | N$_{50m}$ | N$_{100m}$ | Chi-Square | df | *p* |
|-----------|-----------|-----------|------------|------------|-----|-----|
| Length | 250 | 61 | 61 | 4.213 | 2 | 0.12 |
| Mass | 250 | 61 | 61 | 8.645 | 2 | 0.01 |
| Mean thickness | 250 | 61 | 61 | 27.120 | 2 | <0.01 |

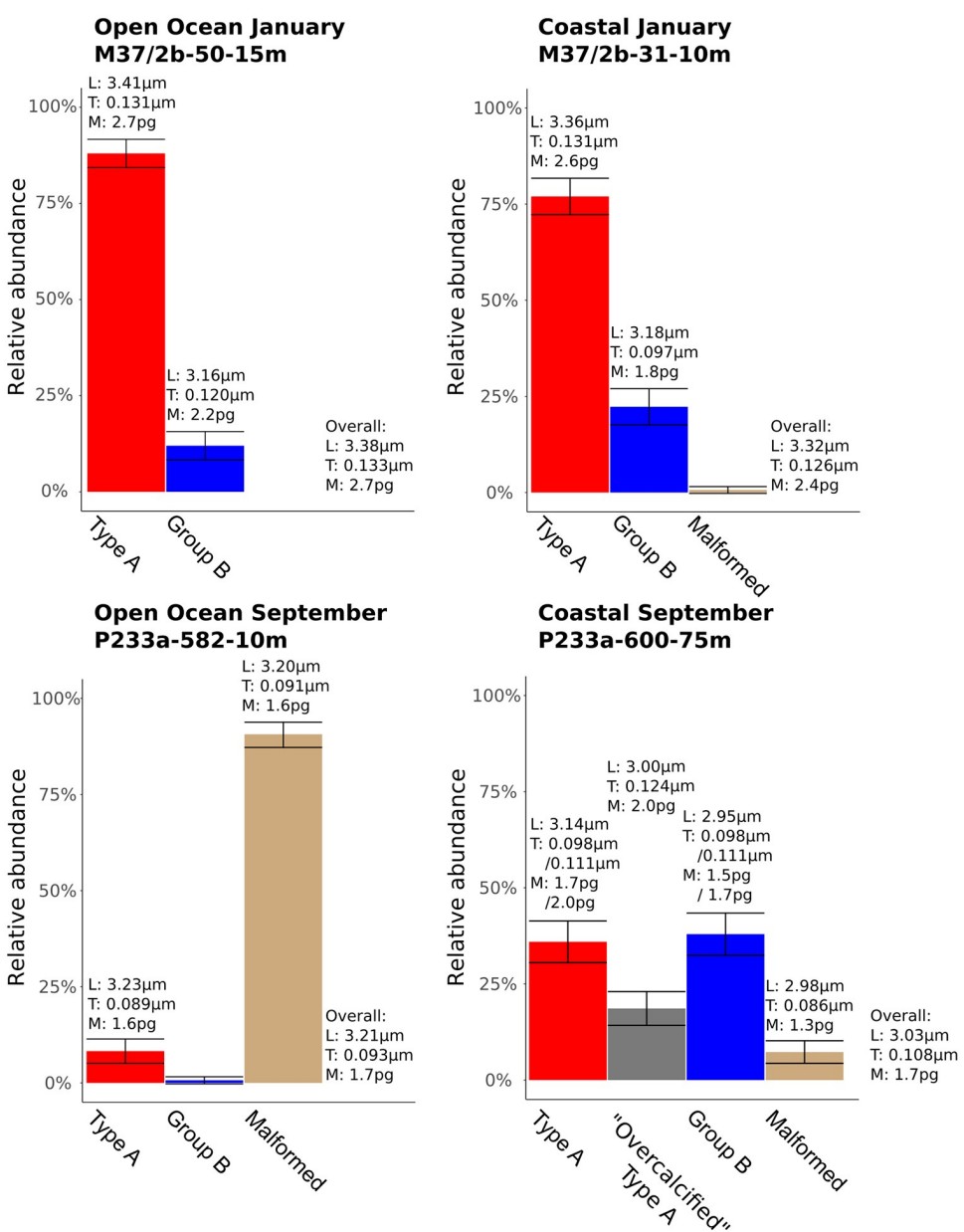

**Fig 11. Relative abundance of *E. huxleyi* morphotypes at each station.** Error bars show margin of error of counts with a 95% confidence level. Overall: Mean value of all *E. huxleyi* coccoliths in a sample. L: Coccolith length. T: Coccolith thickness. M: Coccolith mass.

as estimated from MCMC models and Eq 2 (Figs 11 and 12, Table 7). "Overcalcified" Type A was both heavier and thicker than non-"Overcalcified" Type A morphotypes at the coastal site in September. The same "Overcalcified" Type A coccoliths were, however, on average lighter than the mean measured *E. huxleyi* coccolith mass at both sites in January, which contained no "Overcalcified" Type A (Fig 11, Table 3). In fact, mass approximations done using volumetric calculations (Eq 2) show that Type A coccoliths at the two sites in January were heavier than "Overcalcified" Type A coccoliths in this study. Type A mean coccolith mass in January was 2.7pg at the open ocean site and 2.6pg at the coastal site (Fig 11). Group B coccoliths at the

**Table 7. Markov Chain Monte Carlo (MCMC) analysis results for estimated thickness of *E. huxleyi* morphotypes in each sample.** $\mu$ is the estimated mean thickness, $\sigma$ is the estimated standard deviation of thickness, and CI lower, median, and CI upper shows the 16th, 50th and 84th quartiles of the sampled parameters, giving the 68% Credible Interval for each parameter. Morphotypes: A: Type A; B: Group B; OA: Overcalcified Type A; M: malformed coccoliths. Note that in sample P233a-600-75m the MCMC model could not confidently assign a thickness to Type A or Group B.

| Sample | Morphotype | $\mu$ (µm) | | | $\sigma$ (µm) | | |
|---|---|---|---|---|---|---|---|
| | | CI lower | Median | CI upper | CI lower | Median | CI upper |
| M37/2b-31-10m | A | 0.127 | 0.131 | 0.134 | 0.024 | 0.027 | 0.029 |
| | B | 0.091 | 0.097 | 0.107 | 0.014 | 0.018 | 0.024 |
| M37/2b-50-15m | A | 0.129 | 0.131 | 0.133 | 0.018 | 0.019 | 0.020 |
| | B | 0.110 | 0.120 | 0.130 | 0.018 | 0.019 | 0.020 |
| P233a-600-75m | A or B | 0.094 | 0.098 | 0.103 | 0.008 | 0.011 | 0.015 |
| | A or B | 0.108 | 0.111 | 0.113 | 0.006 | 0.008 | 0.014 |
| | OA | 0.123 | 0.128 | 0.130 | 0.010 | 0.012 | 0.014 |
| | M | 0.083 | 0.086 | 0.090 | 0.009 | 0.011 | 0.013 |
| P233a-582-10m | A | 0.079 | 0.089 | 0.099 | 0.015 | 0.019 | 0.027 |
| | M | 0.089 | 0.091 | 0.094 | 0.013 | 0.014 | 0.016 |

open ocean site in January had similar coccolith mass (2.2pg) as the coastal "Overcalcified" Type A coccoliths found in September (Fig 11).

As a quality control of the morphotype thickness and mass estimations, "Overcalcified" Type A coccolith thickness and mass was also estimated the same way as other morphotypes and compared with the direct LM measurements. Measured "Overcalcified" Type A coccolith thickness (0.124 µm, Table 3) compared well with estimated coccolith thickness from the MCMC model (0.128 µm, Table 7). Similarly, coccolith mass compared well between direct measurements (2.0pg, Table 3) and volumetric model estimation using Eq 2 (2.1pg).

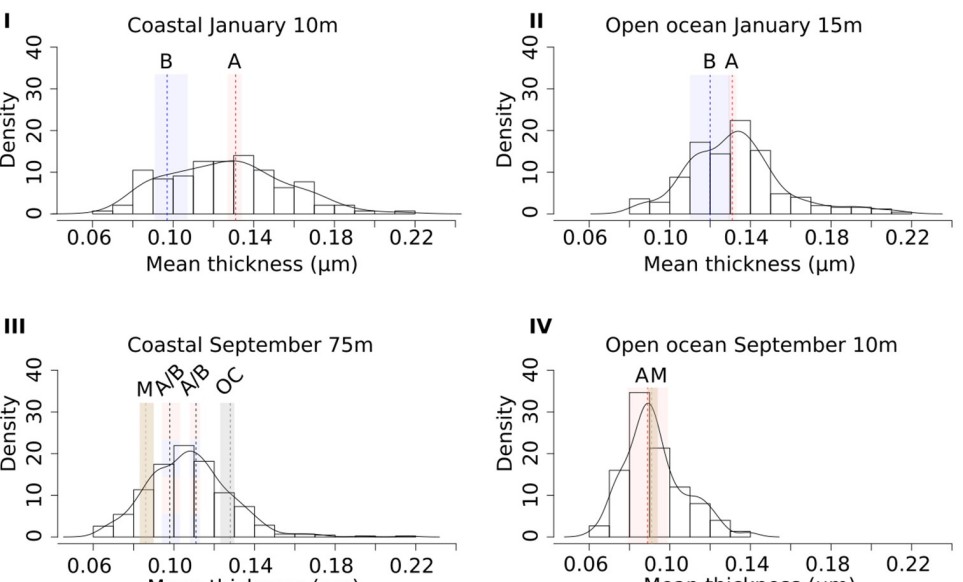

**Fig 12. Histograms of coccolith mean thickness measured at each site.** The height of the bars represent the relative frequency of the bin (density). The black curve shows the density distribution for coccolith thickness in each sample. Coloured vertical bars represent the 68% credible interval for the MCMC-sampled mean coccolith thickness for each morphotype, while the vertical stapled line represent the median sampled mean value. Red bars/lines: Type A. Blue bars/lines: Group B. Grey bar/line: "Overcalcified" Type A. Brown bars/lines: malformed coccoliths.

## MCMC accuracy evaluation

Using a MCMC model for *E. huxleyi* morphotype analysis is a novel approach which allowed for a more complete understanding of intra-morphotypic variation in coccolith morphology and mass in *E. huxleyi*. The accuracy of the MCMC model was evaluated using simulated sample compositions (Table 2). A random number generator was used to create samples with mixtures of two or four morphotypes (two or four components). The mixed values were generated from normal distributions with mean, standard deviation, and weight (relative abundance) set as shown in Table 2. The total number of 16 simulated samples (twelve with two morphotypes and four with four morphotypes) was chosen as a balance between covering a sufficient range of different scenarios and computing time, as each individual MCMC model analysis is time-consuming. Subsequently, MCMC analyses were done based on these simulated thickness distributions and the relative abundances chosen for each simulated morphotype. This approach allowed for the evaluation of the accuracy of the MCMC models on samples resembling the real coccolith samples but where the (simulated) mean thickness is known independently of the MCMC model (S3 Table).

Using the MCMC model it was possible to estimate the mean thickness for the simulated morphotypes with an accuracy of ±0.01 units. The only exceptions were one morphotype in each of the two samples: Group 3: Simulation 3 and Group 4: Simulation 3 (Fig 7). The simulated mean thickness also fell within the 68% credible interval of the MCMC models (for details see Methods section), except for the same two morphotypes in the simulated samples mentioned above (Group 3: Simulation 3 and Group 4: Simulation 3). In total the simulated morphotype mean thickness fell within the 68% credible interval in the case of 95% of the simulated morphotypes (Fig 13). The 68% credible interval thus appears to give a reliable margin of error for the estimated thickness from the MCMC analysis.

The input order for relative abundance was not important for the two-morphotypes MCMC models, but in the four-morphotypes models the morphotype mean thickness values were sorted during analysis to avoid label switching (a phenomenon resulting in the averaging out of individual component mean values—see [75] for details). Because the morphotype mean thickness values in the four-morphotypes models were sorted, the input order for

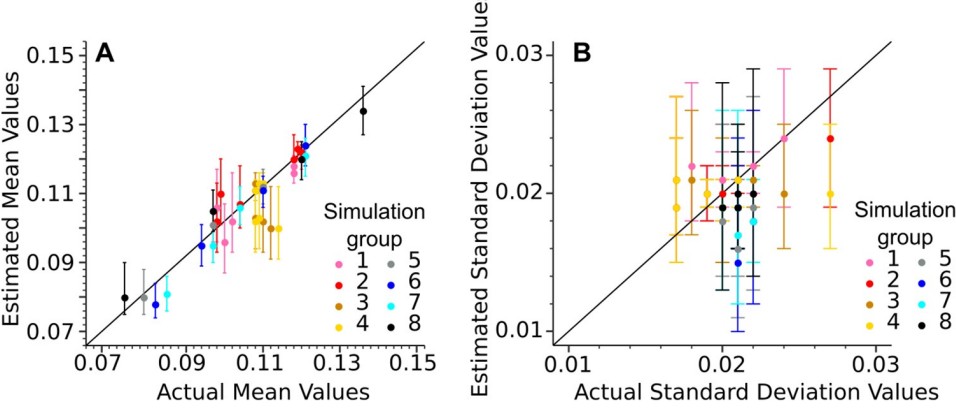

**Fig 13. Scatter plots of actual mean thickness versus mean thickness estimated by the MCMC (A) and actual standard deviation versus standard deviation estimated by the MCMC (B) for each simulated component (simulated morphotype) in the various two- and four-component simulated samples.** Each dot represents the median of each MCMC sampling with the error bars representing the 68% Credible Interval. Each color represents a different simulation group (see subsection on Markov Chain Monte Carlo analysis in the Methods section in text for details for each group).

relative abundance may influence results as the first morphotype, for example, is now both given prior information about its relative abundance (first relative abundance input) and its mean thickness relative to the other morphotypes (least thickness). Tests done on the simulated samples revealed that the MCMC models were able to correct for the influence of the input order.

### Type A central tube width

The ratio of the central tube width to coccolith length (CT:L) was measured in the SEM for 30 Type A coccoliths at each site as a potential measure of calcification. CT:L was equal at both sites in January at 0.07 ±0.01, while CT:L was 0.06 ±0.01 in September at the coastal site and 0.04 ±0.00 in September at the open ocean site. The variation between January and September was statistically significant at both sites, though only at the open ocean site was the difference outside the margin of error.

## Discussion

Relating coccolith morphology to coccolith mass is a challenging task, as only "Overcalcified" Type A can be distinguished in a light microscope (Fig 6). The CPR-method is able to estimate coccolith thickness with high accuracy [61], but differences between most morphotypes, for example Type A and Group B coccoliths, can not be directly measured in a light microscope. Using morphotype relative abundance obtained from SEM images and sample distributions of measured *E. huxleyi* coccolith thickness obtained from the CPR-method as input, a Markov Chain Monte Carlo method was therefore used to get accurate thickness estimations for each *E. huxleyi* morphotype (generally ±0.01 μm or less (Table 7)). Together with these morphotype thickness estimates, morphotype length measurements from SEM images (Table 1) can be used to get reliable estimates of coccolith mass per morphotype (e.g. Eq 2, see also [61]). It was thus possible to obtain insights into differences in coccolith mass between morphotypes, revealing that coccolith morphology and mass of *E. huxleyi* morphotypes vary unsystematically and inconsistently (Figs 11 and 14). "Overcalcified" Type A were, for example, heavier than Type A coccoliths in some samples, but lighter than Type A coccoliths in other samples. Similarly, "Overcalcified" Type A were either heavier or same mass as Group B coccoliths. In fact, no morphotype was consistently heavier or lighter than another morphotype when compared across different samples (Fig 14A). *E. huxleyi* cell density also varied between samples, resulting in significantly different calcite production by *E. huxleyi* even when relative abundance of morphotypes were similar (Fig 15B). In effect, the relative abundance of an individual morphotype may not be used as a proxy for either mean coccolith mass or *E. huxleyi* calcite production (Fig 15).

   These results are in contrast with studies that have relied on SEM imaging and an a priori assumption of coccolith mass differences between morphotypes to make inferences on the impacts of ocean acidification on *E. huxleyi* calcite production (e.g. [32, 33, 37]). These studies solely rely on the distal shield appearance of morphotypes to make inferences about relative coccolith mass differences between different morphotypes, but other aspects of the coccolith also appear to be significant. The greater thickness of Type A coccoliths in January than "Overcalcified" Type A in September may, for example, be due to other factors influencing coccolith thickness and mass. Proximal shield closure or central tube height, which are not obvious from SEM images, might be equally important. This could also explain the similar PIC per cell in the "Overcalcified" Type A *E. huxleyi* strain C352 in [39] compared to normal Type A *E. huxleyi* strains in the same study.

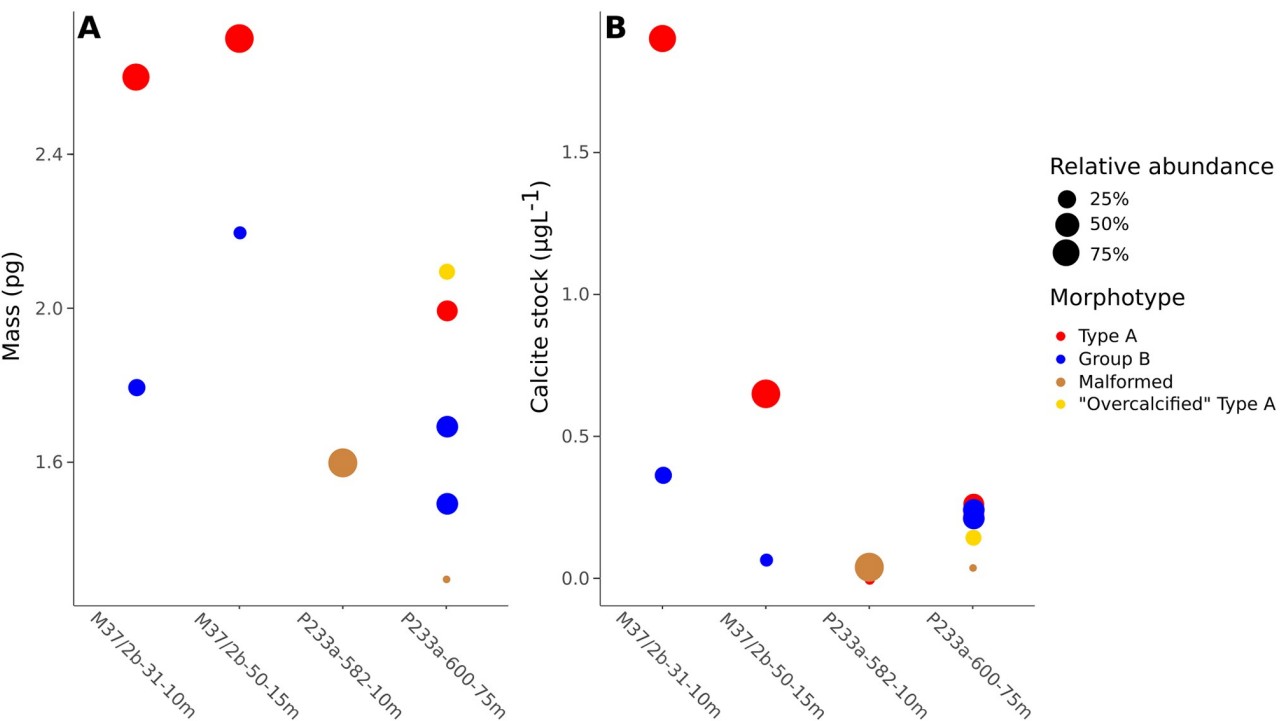

**Fig 14.** A: Mean coccolith mass of individual *E. huxleyi* morphotypes estimated using a volumetric model (Eq 2). B: Calcite production by individual *E. huxleyi* morphotypes. Calcite concentration was estimated using measured coccolith mass in A, cell density, and assuming 23 coccoliths per cell. Colour of circle denotes the morphotype, while the size of the circle indicates the relative abundance of the morphotype in each sample. Note that in sample P233a-600-75m Type A and Group B coccoliths had each two possible mean coccolith thickness values obtained from the MCMC model. Therefore, two alternative coccolith mass and calcite concentration estimates are given for each of these morphotypes.

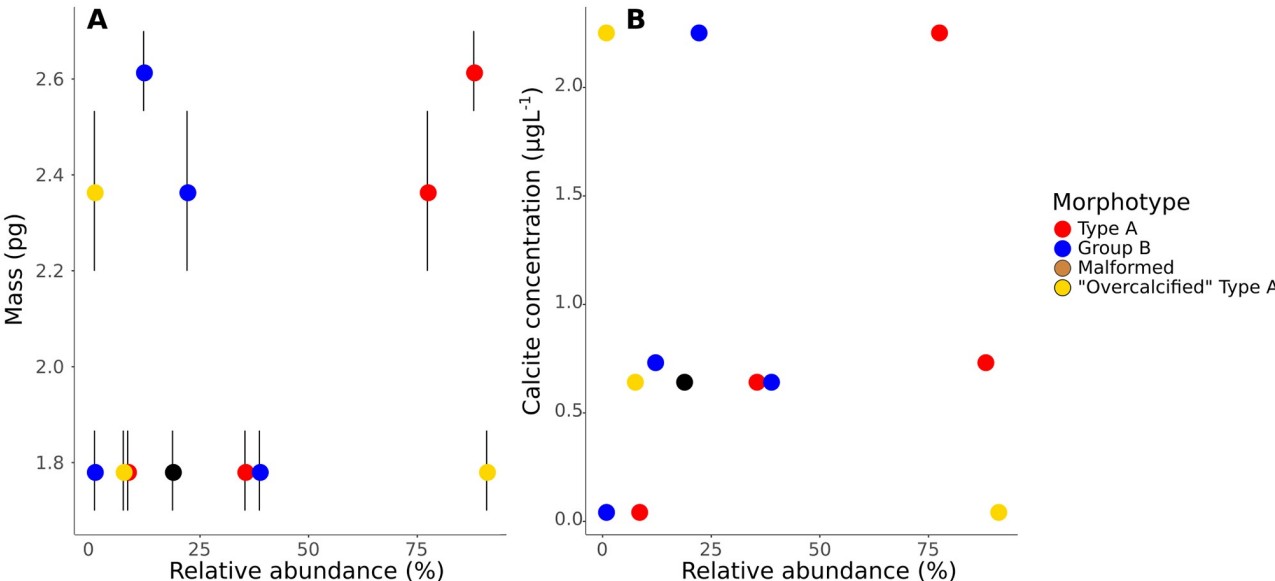

**Fig 15.** A: *E. huxleyi* morphotype relative abundance versus mean coccolith mass. Mean coccolith mass was measured directly from LM images using the CPR-method. Error bars represent the 95% confidence interval of the mean. B: *E. huxleyi* morphotype relative abundance versus estimated calcite concentration of *E. huxleyi*. Calcite concentration was estimated using measured coccolith mass in A, cell density, and assuming 23 coccoliths per cell. Colour of circle denotes the morphotype.

In summary, the term "Overcalcified" appears to be a poor term with potentially misleading implications. Referring to Group B coccoliths as "lightly calcified" likewise appears to be misleading. Group B coccoliths at the open ocean site in January appear to be thicker than Type A in September at the coastal site (Table 7), despite similar coccolith size (Table 1). Referring to Group B coccoliths as lightly or less calcified compared to Type A is therefore unjustified because it is based only on the distal shield appearance.

## Malformed coccoliths

Several studies have linked ocean acidification to increased malformation in culture experiments (e.g. [5, 76, 77]), though [78] reported that this response is strain-specific in *E. huxleyi*. Furthermore, in recent culture experiments calcite content per cell did not decrease in *E. huxleyi* even when more than 95% of coccoliths were malformed [43, 44]. In line with the findings of [43, 44], coccolith mass in the present study was the same at both study sites in September, despite the high (>90%) proportion of malformed coccoliths at the open ocean site (Table 3). Malformed coccoliths appear therefore to be a poor indicator of coccolith mass, in the same way that different morphotypes are.

## Central tube width

Type A central tube width have been linked to different degrees of calcification in *E. huxleyi* [30, 40]. In the present study CT:L ratio followed Type A coccolith mass, as both decreased together from January to September at both sites. Central tube width is a factor likely contributing to coccolith mass, but CT:L alone does not seem to explain the full variation in coccolith mass seen within Type A in the present study. For example, the CT:L ratio decreased by ~15% at the coastal site from January to September, but coccolith mass decreased by 23-35% (Fig 11). Moreover, Type A calcite concentration decreased by ~86-88%, showing that variation in Type A CT:L is not a suitable indicator of *E. huxleyi* calcite production either.

## MCMC assumptions

The MCMC model relies on the assumption that the samples can be described by two or four morphotypes with approximately normal distributions of morphological parameters. The number of components/morphotypes in the model reflects the number of morphotypes counted in the SEM, with the exception of very rare morphotypes (<1% relative abundance). Previous studies have reported several different morphotypes within both Type A [40] and Group B (e.g. [34, 35]), and the presence of different sub-varieties of these morphotypes may invalidate this assumption. However, there is no evidence of bimodality in the Type A or Group B coccoliths, suggesting they are well represented as one morphological unit in the present study. While the main difference between Group B morphotypes is size [57], the length of Group B coccoliths in this study is normally distributed in all samples according to a Shapiro-Wilk's test. Group B coccoliths in the present study therefore seem to represent a single morphotype, at least within an individual sample. Shapiro-Wilk's tests also revealed that coccolith length and CT:L is normally distributed in Type A, with the exceptions of Type A coccolith length in the Open ocean January sample (Shapiro Wilk's $p = 0.048$) and Type A CT:L in the Coastal January sample (Shapiro Wilk's $p = 0.013$). These are due to the influence of two or one outliers, respectively. If the outliers are removed, the Shapiro Wilk's $p > 0.05$, and the distributions are normal as well. Therefore, Type A coccolith length and CT:L measurements are considered to be normally distributed, and Type A coccoliths seem to represent one normally distributed unit within each individual sample.

A potential exception to the assumptions is seen in the Open ocean sample in September. The three Markov chains used for the parameter sampling by the MCMC model appeared to converge in this sample (S3 Fig), but the mean values sampled by the model were often obviously wrong for the Type A component in the Open ocean September sample. They were often smaller than the lowest measured value, and sometimes even negative. This was not seen for the other models, and may mean that the two components chosen for this sample model are wrong. Either malformed and non-malformed coccoliths in this sample may not be separated from each other in terms of thickness, or Type A coccoliths in this sample are too rare to contribute significantly to the sample thickness distribution. Unfortunately, it was not possible to obtain more coccoliths even after preparing a second slide from the same sample, because the sample is so sparse in coccoliths.

## Comparison with other studies

Other studies on morphotype-specific differences in *E. huxleyi* coccolith mass have combined relative abundance estimates with coccolith mass estimates obtained from their birefringence in a polarized light microscope (e.g. [8, 40]). These studies did not use MCMC analysis or other tools to estimate coccolith mass per morphotype, but rather correlated changes in *E. huxleyi* (or Noelrhabdaceae) coccolith mass with changes in morphotype relative abundance. However, this approach appears to be not suitable to disentangle the effects of changes in relative abundance from intra-morphotypic changes in coccolith mass. For example, in the present study, the ~25-36% decrease in Type A coccolith mass at the coastal site from January to September is not obvious from looking at the changes in relative abundance over the same time period (Fig 14). Moreover, while Type A coccolith size and thickness both decreased at the coastal site from January to September, Group B coccoliths decreased in size but not in thickness (Tables 1 and 7). Similarly, Group B coccoliths were thicker at the open ocean site in January compared to the coastal site, while Type A coccolith thickness did not change. The complex mass variation patterns among morphotypes in the present study is consistent with culture studies showing that environmental responses in *E. huxleyi* are both strain-specific [11, 78, 79] and morphotype-specific [39, 41, 80, 81], and is unlikely to be unique to the Canary Islands region. On the other hand, the present study appears to contradict, for example, [8], who reported that *E. huxleyi* coccolith mass was driven by variation in *E. huxleyi* morphotype relative abundance. The reason for this contradiction lies in part in the applied method of [8], which has been shown to have significant flaws detailed in [61, 82–86]. In addition, [8] measured only coccolith mass at a family level (*Gephyrocapsa oceanica* plus *E. huxleyi*) without analyzing the coccolith mass of individual *E. huxleyi* morphotypes. The present study, however, demonstrated that coccolith mass variation within individual morphotypes may be an important factor in explaining coccolith mass differences between samples.

Another widely used approach to understand intraspecific coccolith mass variation in *E. huxleyi* has been to estimate coccolith mass from SEM images using the $k_s$ model (e.g. [26, 28, 87]). In this model mass is estimated using measured coccolith length and a species or morphotype-specific $k_s$ value. However, the $k_s$ values for both Type A and Type B were defined based on a very limited number of coccoliths (four and three, respectively [26]). Moreover, a recent culture study found that coccolith thickness did not vary proportionally with coccolith length in several Type A *E. huxleyi* strains [88]. Similarly, in the present study Group B coccoliths varied in thickness, but not length, between the coastal and open ocean site in January. Intra-morphotypic coccolith mass variation is purely a reflection of variation in coccolith length in the $k_s$ model. Therefore the model may present an incomplete picture of the true variation in coccolith mass.

### Relationship between *E. huxleyi* coccolith mass and calcite concentration

The present study demonstrated that the relationship between coccolith mass and *E. huxleyi* morphotype relative abundance is not straightforward. The relationship between coccolith mass and the total amount of calcite produced by *E. huxleyi* appears to be also not straightforward, as illustrated by the estimated *E. huxleyi* calcite concentrations. Please note that the calcite concentration estimations presented here are not intended as accurate representations of calcite produced by *E. huxleyi* at each site. Instead, the intention of these estimations is to highlight the importance of factors other than simply coccolith mass when determining coccolithophore calcite production. Considering coccolith mass alone, and assuming coccolith mass is closely related to coccolithophore calcite mass [8] (i.e. the number of coccoliths per coccolithophore cell is similar in different samples), one might be tempted to conclude that calcite production is similar at the two sites in the present study (Table 5). However, when cell density is taken into account, it is obvious that calcite production at the coastal site is much greater than at the open ocean site because of the differences in cell density (Fig 10, Table 5). At the same time, the greater *E. huxleyi* coccolith mass at the open ocean site in January gives greater calcite production compared to the coastal site in September, despite the lower cell density (Table 5). Neither cell density or coccolith mass should be ignored for accurate calcite production estimation in the ocean.

Accurate calcite production estimates require accurate knowledge of the number of coccoliths per cell as well. In the present study, 10, 23, and 48 coccoliths per cell (taken from [73] and [74]) were used to illustrate the importance of coccolith numbers for overall calcite concentration estimation. [89] reported even greater variation in number of coccoliths per cell from samples collected at different cruises along the Portuguese coast in the North Atlantic, where they varied from 4 to 135 coccoliths per cell (excluding samples where no coccospheres/cells were seen). Furthermore, *E. huxleyi* is known to shed coccoliths during growth, particularly during bloom conditions [20], which may also impact calcite concentration estimations.

Following the same reasoning, relative contributions of individual morphotypes to *E. huxleyi* calcite production does not only depend on the coccolith mass of the individual morphotypes, but also on the cell density and coccolith production of the morphotypes. Type B coccoliths have, for example, been reported to produce more coccoliths per cell than Type A in a culture study, accompanied by greater calcite production [80]. Whether studying calcite production at species or sub-species level, neither cell density nor coccolith mass and coccoliths per cell should be ignored for accurate estimates.

## Conclusion

This study combined thickness measurements from the CPR-method and *E. huxleyi* morphotype relative abundance from SEM images with MCMC analysis to estimate coccolith thickness and mass of individual *E. huxleyi* morphotypes in different sampling sites and seasons near the Canary Islands. This analysis revealed that thickness and/or mass can not be used to characterize a specific *E. huxleyi* morphotype nor to describe the relations between different morphotypes (e.g. characterizing "Overcalcified" Type A coccoliths as heavier than Group B coccoliths). Coccolith length and thickness of morphotypes vary both seasonally and regionally, so that changes in relative abundance of morphotypes can not be used as a proxy for either *E. huxleyi* coccolith mass or calcite production. Furthermore, this study revealed that the terms "Overcalcified" and "lightly calcified" are misleading; "Overcalcified" Type A coccoliths in this study were lighter and thinner than some "normally calcified" Type A coccoliths and similar in thickness to some "lightly calcified" Group B coccoliths. This study highlights the

challenges in trying to estimate coccolithophore calcite production from coccolith habitus, and uncritically using coccolith habitus as a proxy for calcification appears to be ill-advised.

## Supporting information

**S1 Fig. Traceplot for the MCMC model for the open ocean site in January.** Blue, green and red lines each represent an individual chain for the sampling.
(TIFF)

**S2 Fig. Traceplot for the MCMC model for the coastal site in January.** Blue, green and red lines each represent an individual chain for the sampling.
(TIFF)

**S3 Fig. Traceplot for the MCMC model for the open ocean site in September.** Blue, green and red lines each represent an individual chain for the sampling.
(TIFF)

**S4 Fig. Traceplot for the MCMC model for the coastal site in September.** Blue, green and red lines each represent an individual chain for the sampling.
(TIFF)

**S1 Table. SEM Counts and measurements from the open ocean and coastal sites in January and September 1997.** A: Type A; B: Group B; OA: "Overcalcified" Type A; M: Malformed.
(CSV)

**S2 Table. LM coccolith measurements from samples used in the study.** Images were captured in RAW and converted to TIFF in sRGB with 2.2 gamma. Coccoliths were segmented from the image background using a Canny-Deriche edge detection algorithm.
(CSV)

**S3 Table. Mean thickness, standard deviation, and relative abundance of simulated samples generated with a random number generator.** RA: Relative abundance; SD: Standard deviation. Note that samples are simulated with no units.
(CSV)

## Acknowledgments

This work was possible thanks to the efficient work at sea of the captain and crew of the research vessels F.S. METEOR and F. S. POSEIDON. Thanks go to Tim Rodgers and two anonymous reviewers for their helpful comments.

## Author Contributions

**Conceptualization:** Simen Alexander Linge Johnsen, Jörg Bollmann.

**Formal analysis:** Simen Alexander Linge Johnsen.

**Funding acquisition:** Jörg Bollmann.

**Investigation:** Simen Alexander Linge Johnsen.

**Methodology:** Simen Alexander Linge Johnsen, Jörg Bollmann.

**Project administration:** Simen Alexander Linge Johnsen.

**Resources:** Jörg Bollmann.

**Software:** Simen Alexander Linge Johnsen.

**Supervision:** Jörg Bollmann.

**Validation:** Simen Alexander Linge Johnsen, Jörg Bollmann.

**Visualization:** Simen Alexander Linge Johnsen.

**Writing – original draft:** Simen Alexander Linge Johnsen.

**Writing – review & editing:** Jörg Bollmann.

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
