## [Decision Letter · Decision Letter 0]

1 Oct 2019

PONE-D-19-23241

Coccolith mass and morphology of different Emiliania huxleyi morphotypes

PLOS ONE

Dear Dr. Linge Johnsen,

Thank you for submitting your manuscript to PLOS ONE. After careful consideration, we feel that it has merit but does not fully meet PLOS ONE’s publication criteria as it currently stands. Therefore, we invite you to submit a revised version of the manuscript that addresses the points raised during the review process.

We would appreciate receiving your revised manuscript by Nov 15 2019 11:59PM. To enhance the reproducibility of your results, we recommend that if applicable you deposit your laboratory protocols in protocols.io, where a protocol can be assigned its own identifier (DOI) such that it can be cited independently in the future. For instructions see: http://journals.plos.org/plosone/s/submission-guidelines#loc-laboratory-protocols

We look forward to receiving your revised manuscript.

Kind regards,

Christopher Edward Cornwall, Ph.D.

Academic Editor

PLOS ONE

**Journal Requirements:**

**Additional Editor Comments (if provided):**

I have received back two referee reports. Both indicate that major revisions would be required before this manuscript could be considered for publication. The reviewers' comments are straight forward, but both point out that the methods are currently unclear. Please address all of these concerns if you wish to submit a revised manuscript for further consideration.

**Comments to the Author**

1. Is the manuscript technically sound, and do the data support the conclusions?

Reviewer #1: Partly

Reviewer #2: Partly

2. Has the statistical analysis been performed appropriately and rigorously? 

Reviewer #1: I Don't Know

Reviewer #2: Yes

3. Have the authors made all data underlying the findings in their manuscript fully available?

Reviewer #1: Yes

Reviewer #2: Yes

4. Is the manuscript presented in an intelligible fashion and written in standard English?

Reviewer #1: Yes

Reviewer #2: Yes

5. Review Comments to the Author

Reviewer #1: Emiliania huxleyi without doubt is the best studied calcifying nanoplankton. However from studying different global strains/populations it has become clear this organism represents an enormous physiological and genetic diversity, potentially deserving at least discrimination of several varieties (Medlin 1994). As a first approximation researchers therefore have tried to discriminate between "morphotypes" such as A, A overcalcified, B/C etc. These ecotypes roughly represent differing sensitivities to ocean acidification (Muller et al. 2005 MEPS) and , at least in the Southern Ocean, have different temperature tolerance and ranges (Cubillos 2007, MEPS) and light responses. The present work makes a valid point that over time, these morphotypes , sometimes carelessly called "calcification morphotypes", have become equated with calcite density. This is something that requires more careful measurements using a combination of SEM measurements, birefringence polarising microscopy (Beaufort method), but also (not conducted here, because working with field samples, not clonal cultures) coulter counter sizing of coccosphere diameter and number of coccoliths and sizes.

The present work on a mixed assemblage from the Canary Islands claims that A, A overcalcified, B did not significantly differ in mass and thickness. This conclusion contradicts previous work by Young & Ziveri 2000. Deep-Sea Research 47,1679-1700 (2000),Poulton et al. 2013, Global Biochemical Cycles 27, 1023-1033, Poulton et al. 2011, Mar. Ecol.-Prog. Ser., 443, 1–17, Charalampopoulou et al. 2016, Biogeosciences, 13, 5917–5935; Beaufort et al. 2011. Nature. etc

The reasons for this discrepancy need to be carefully explored. Either the present authors' conclusion (no differences between morphotypes and mass/thickness) are correct but relate only to the particular Canary Island coccolithophorid population studied (which I find hard to believe). OR , this is due to differences in methodology or concepts used. The criticisms against the Beaufort birefringence method [line 486] are too easily raised however and lack detail. Another complication is that different morphotypes have different coccosphere diameters and produce different numbers of coccoliths of differing size as well as calcite mass. Note that coccosphere diameter is not considered here and number of liths is largely ignored as well (line 301, only guessed ). Thickness and mass (line 23) cannot be estimated from SEM and were estimated here from LM and a Markov Chain Monte Carlo analysis. I must admit that I have difficulty following the assumptions using this approach, but whatever, this needs to be better explained.

The authors raise a good point of warning against carelessly equating morphotypes with calcification fluxes, BUT their results contradict a significant body of literature, and they need to better explain differences in concepts (not coccosphere, not number of liths), but also of methods (MCMC) and materials used.

Reviewer #2: This study mainly combined the CPR-method and MCMC analysis to estimate the coccolith thickness and calcite mass of two different Emiliania huxleyi morphotypes sampled off the northwestern African coast. The results suggest that the coccolith length and thickness did not necessarily correlate with the characteristics of different E. huxleyi morphotypes. Therefore, the study provides some insights on further understanding the complex implication of the existence of different morphotypes of the cosmopolitan coccolithophore species E. huxleyi on calcite production. However, some of the important details are missing, the results were poorly discussed and the structure of the discussion section needs to be reorganized. I would suggest the manuscript resubmitted for review after some severe revisions.

My specific comments are listed below.

Abstract: The present abstract is poorly written. It seems lacking important information of the study. Please add some brief description of the methodology (i.e. how the morphotypes, mass and thickness of coccoliths were examined), the major results/findings and the oceanographic implications of the study.

Introduction

Line 22: I think the word “organic” should be revised to “inorganic”, that is the carbon pool produced by calcification.

Lines 26-28: It says “sever studies…”, however, there is only one reference [8] cited here.

Oceanographic settings:

Lines 84-86: Please specify how the primary productivity was measured.

Materials and Methods:

Lines 93-96: Please specify how the sampling depths of this study were determined. It is mentioned in the text that “ the depths… were chosen based on E. huxleyi cell densities”. Were they the depths where the maximum E. huxleyi cell abundances observed? Were there any other coccolithophore species observed? According to the cited reference [45] (Bollmann J, Cortés MY. Distribution of Living Coccolithophores North of the Canary Islands: Vertical Seasonal and Interannual Variations. In: American Geophysical Union, Fall Meeting 2007; 2007. p. OS11A–0189), the dominant coccolithophore species in the upper photic zone was Gephyocapsa ericsonii.

Lines 103-108: Please specify how the Type A coccoliths were defined and grouped.

Results:

Lines 219-222: This paragraph should be moved to the “Materials and Methods” section.

Lines 251- 256: Please add some details on the specific differences in thickness between the open ocean and coastal sampling sites.

Discussion:

Line 317: “This study characterizes E. huxleyi morphotyples using coccolith length and thickness”: I think this is confusing, since later in lines 341-344, the authors mentioned that the morphotypes were somehow determined by the appearance of the distal shield elements.

Lines 342-343: Again, please specify how the “overcalcified” and “normal” Type A coccoliths were defined and separated in detail. From Fig. 13, I only found two groups of different morphotypes (Type A with closed central area in red and Group B with an open central area in black). How was the conclusion “this study revealed that despite their appearance, “overcalcified” Type A coccoliths are not necessarily heavier or thicker than “normal” Type A coccoliths” drawn based on this figure?

Lines 355-357: The current expression is confusing. Similar thickness and length don’t necessarily imply similar coccolith mass.

Lines 362-363: It is mentioned here that “variable proximal shield slit closure, width of the central tube, thickness of distal shield elements and the size of the proximal shield related to the distal shield” may all contribute the “degree of calcification” of coccolith. However, the authors used a way too simplified model (Eq. 3) to estimate the coccolith mass by only considering the length, width and thickness of coccolith. Therefore, this would have generated rather biased results and further conclusions.

Lines 393-426: This paragraph should be moved to the “results” section.

Lines 463-467: How would these exceptions affect the outcome of the analysis?

Line 473: What is the assumption in reference #31? Please provide some details.

Lines 494-498: The expression is confusing. How can you extrapolate the trend for coastal site based on the observations for open ocean site?

In general, the discussion is rather vague at this stage. For example, were there any potential effects of the environmental conditions (such as seawater carbonate chemistry, nutrient concentrations, temperature and salinity) on the coccolith mass and calcite stock? Was there any difference observed between the coastal and open ocean site? These are important factors controlling the calcification process in coccolithophores and therefore will regulate the coccolith mass and calcite production of the cells. I think the authors should take these factors into consideration.

6. PLOS authors have the option to publish the peer review history of their article (what does this mean?). If published, this will include your full peer review and any attached files.

Reviewer #1: No

Reviewer #2: No

---

## [Author Response · Author response to Decision Letter 0]

29 Nov 2019

Response to reviewers

Reviewer #1: "Emiliania huxleyi without doubt is the best studied calcifying nanoplankton. However from studying different global strains/populations it has become clear this organism represents an enormous physiological and genetic diversity, potentially deserving at least discrimination of several varieties (Medlin 1994). As a first approximation researchers therefore have tried to discriminate between "morphotypes" such as A, A overcalcified, B/C etc. These ecotypes roughly represent differing sensitivities to ocean acidification (Muller et al. 2005 MEPS) and , at least in the Southern Ocean, have different temperature tolerance and ranges (Cubillos 2007, MEPS) and light responses. The present work makes a valid point that over time, these morphotypes , sometimes carelessly called "calcification morphotypes", have become equated with calcite density. This is something that requires more careful measurements using a combination of SEM measurements, birefringence polarising microscopy (Beaufort method), but also (not conducted here, because working with field samples, not clonal cultures) coulter counter sizing of coccosphere diameter and number of coccoliths and sizes.

The present work on a mixed assemblage from the Canary Islands claims that A, A overcalcified, B did not significantly differ in mass and thickness. This conclusion contradicts previous work by Young & Ziveri 2000. Deep-Sea Research 47,1679-1700 (2000),Poulton et al. 2013, Global Biochemical Cycles 27, 1023-1033, Poulton et al. 2011, Mar. Ecol.-Prog. Ser., 443, 1–17, Charalampopoulou et al. 2016, Biogeosciences, 13, 5917–5935; Beaufort et al. 2011. Nature. etc

The reasons for this discrepancy need to be carefully explored. Either the present authors' conclusion (no differences between morphotypes and mass/thickness) are correct but relate only to the particular Canary Island coccolithophorid population studied (which I find hard to believe). OR , this is due to differences in methodology or concepts used. The criticisms against the Beaufort birefringence method [line 486] are too easily raised however and lack detail. "

We agree with the reviewer that discrepancies between this study and other studies is not likely to be due to geographical peculiarities, but is rather methodological. Bollmann (2013,2014) outlined in detail the issues using the method proposed by Beaufort (2005). In addition, most studies have not captured a complete picture of variation within individual morphotypes (e.g. Young and Ziveri (2000), Beaufort et al. (2011), D’Amario et al. (2018). We have revised the discussion section to better explain this point.

"Another complication is that different morphotypes have different coccosphere diameters and produce different numbers of coccoliths of differing size as well as calcite mass. Note that coccosphere diameter is not considered here and number of liths is largely ignored as well (line 301, only guessed). "

Our main goal was to estimate the mass of single coccoliths of different morphotypes. However, in order to demonstrate the accumulated effect when coccoliths per cell are used for total calcite mass estimates of coccospheres we did some simple calculations based on published data on coccoliths per cell.

"Thickness and mass (line 23) cannot be estimated from SEM and were estimated here from LM and a Markov Chain Monte Carlo analysis. I must admit that I have difficulty following the assumptions using this approach, but whatever, this needs to be better explained."

We added a paragraph on MCMC assumptions in the Methods section and rewrote parts of the Discussions section on the MCMC assumptions to better explain the methodology. 

"The authors raise a good point of warning against carelessly equating morphotypes with calcification fluxes, BUT their results contradict a significant body of literature, and they need to better explain differences in concepts (not coccosphere, not number of liths), but also of methods (MCMC) and materials used."

We modified the Discussions section and added more elaborated comparisons with other studies.

Reviewer #2: "This study mainly combined the CPR-method and MCMC analysis to estimate the coccolith thickness and calcite mass of two different Emiliania huxleyi morphotypes sampled off the northwestern African coast. The results suggest that the coccolith length and thickness did not necessarily correlate with the characteristics of different E. huxleyi morphotypes. Therefore, the study provides some insights on further understanding the complex implication of the existence of different morphotypes of the cosmopolitan coccolithophore species E. huxleyi on calcite production. However, some of the important details are missing, the results were poorly discussed and the structure of the discussion section needs to be reorganized. I would suggest the manuscript resubmitted for review after some severe revisions.

My specific comments are listed below.

Abstract: The present abstract is poorly written. It seems lacking important information of the study. Please add some brief description of the methodology (i.e. how the morphotypes, mass and thickness of coccoliths were examined), the major results/findings and the oceanographic implications of the study."

The abstract was rewritten and expanded.

"Introduction

Line 22: I think the word “organic” should be revised to “inorganic”, that is the carbon pool produced by calcification."

The word should be organic. The sentence was slightly revised to clarify.

"Lines 26-28: It says “sever studies…”, however, there is only one reference [8] cited here."

We added more references.

"Oceanographic settings:

Lines 84-86: Please specify how the primary productivity was measured."

We did not measure primary productivity. It is stated in the cited reference. 

"Materials and Methods:

Lines 93-96: Please specify how the sampling depths of this study were determined. It is mentioned in the text that “ the depths… were chosen based on E. huxleyi cell densities”. "

Correct, highest cell densities of EHUX to obtain sufficient specimens for measurements. Two sentences were added to the materials section to clarify.

"Were they the depths where the maximum E. huxleyi cell abundances observed?"

The sample selection was mainly based on cell density, plus two other depths at one site to check whether depth has a significant effect on coccolith mass. 

"Were there any other coccolithophore species observed? "

Yes (see Bollmann and Cortes (2007) and Abrantes et al. (2002) cited in the paper), but the presence of other coccolithophores species is not relevant to this study.

"According to the cited reference [45] (Bollmann J, Cortés MY. Distribution of Living Coccolithophores North of the Canary Islands: Vertical Seasonal and Interannual Variations. In: American Geophysical Union, Fall Meeting 2007; 2007. p. OS11A–0189), the dominant coccolithophore species in the upper photic zone was Gephyocapsa ericsonii."

That is correct. 

"Lines 103-108: Please specify how the Type A coccoliths were defined and grouped."

They were defined according to Young et al. (2003). The text was slightly revised to clarify.

"Results:

Lines 219-222: This paragraph should be moved to the “Materials and Methods” section."

The start of the Results section was rewritten and restructured.

"Lines 251- 256: Please add some details on the specific differences in thickness between the open ocean and coastal sampling sites."

Details on the thickness difference between the coastal and open ocean site in January were added. 

"Discussion:

Line 317: “This study characterizes E. huxleyi morphotyples using coccolith length and thickness”: I think this is confusing, since later in lines 341-344, the authors mentioned that the morphotypes were somehow determined by the appearance of the distal shield elements."

The Discussion section has been revised. 

"Lines 342-343: Again, please specify how the “overcalcified” and “normal” Type A coccoliths were defined and separated in detail. From Fig. 13, I only found two groups of different morphotypes (Type A with closed central area in red and Group B with an open central area in black). How was the conclusion “this study revealed that despite their appearance, “overcalcified” Type A coccoliths are not necessarily heavier or thicker than “normal” Type A coccoliths” drawn based on this figure?"

Figure 13 showed “Overcalcified” Type A in red and all other, non-”Overcalcified” Type A morphotypes (i.e. Type A, Group B, and malformed) in black. After some consideration, Figure 13 doesn’t add much to the discussion, and here seems to cause some misunderstanding. It was therefore removed, and some relevant information instead added to Table 4. The relevant section in the Materials and Methods section was also slightly revised to clarify the identification.

"Lines 355-357: The current expression is confusing. Similar thickness and length don’t necessarily imply similar coccolith mass."

We are not sure what the reviewer means here. Coccolith mass is a function of volume and density of the coccolith. It can be assumed that coccolith density does not vary (as they consist of only one mineral, calcite). Volume varies depending on (mean) thickness and area. The area can be simply calculated from the coccolith length and width, assuming an elliptical shape. Nevertheless, the confusing sentence was changed in the Discussion revision.

"Lines 362-363: It is mentioned here that “variable proximal shield slit closure, width of the central tube, thickness of distal shield elements and the size of the proximal shield related to the distal shield” may all contribute the “degree of calcification” of coccolith. However, the authors used a way too simplified model (Eq. 3) to estimate the coccolith mass by only considering the length, width and thickness of coccolith. Therefore, this would have generated rather biased results and further conclusions."

The average thickness obtained from retardation measurements reflects the variation of proximal shield slit closure, width of the central tube, thickness of distal shield elements and the size of the proximal shield related to the distal shield and therefore the results are not biased.

"Lines 393-426: This paragraph should be moved to the “results” section."

The paragraph was moved to the Results section as suggested.

"Lines 463-467: How would these exceptions affect the outcome of the analysis?"

If these indications imply that the morphotypes are not picked from a normally distributed population, it might affect the results. However, the exceptions appear to be due to the influence of one or two outliers, and the p-values are relatively large as well (above 0.01), so we argue there is not strong enough evidence that the morphotypes do not represent normally distributed populations. 

"Line 473: What is the assumption in reference #31? Please provide some details."

The assumption is that Overcalcified Type A is systematically heavier/more calcified than Type A, which this study showed is not a valid assumption. The text has been revised/changed in the Discussion section revision.

"Lines 494-498: The expression is confusing. How can you extrapolate the trend for coastal site based on the observations for open ocean site?"

We clarified that there is no systematic difference in mass of coccoliths with depth . 

"In general, the discussion is rather vague at this stage. For example, were there any potential effects of the environmental conditions (such as seawater carbonate chemistry, nutrient concentrations, temperature and salinity) on the coccolith mass and calcite stock? "

There might be several environmental factors influencing coccolith mass and morphotype abundance. However, it was not the goal of our study to analyse these factors. We looked at the mass and relative abundance of different morphotypes to the test whether morphotypes can be related to overall calcification. The driving factors will the investigated in future studies and is outside of the scope of the present study.

"Was there any difference observed between the coastal and open ocean site? "

Yes there were. See introduction/oceanographic settings, plus Fig 2. 

"These are important factors controlling the calcification process in coccolithophores and therefore will regulate the coccolith mass and calcite production of the cells. I think the authors should take these factors into consideration."

The goal of the study was to investigate whether different morphotypes indicate different degrees of calcification/coccolith mass. The driving environmental actors behind the coccolith mass/calcification is a separate issue that requires detailed data on the carbon chemistry of the water that are not available for all samples used in this study.

---

## [Decision Letter · Decision Letter 1]

10 Jan 2020

PONE-D-19-23241R1

Coccolith mass and morphology of different Emiliania huxleyi morphotypes

PLOS ONE

Dear Dr. Linge Johnsen,

Thank you for submitting your manuscript to PLOS ONE. After careful consideration, we feel that it has merit but does not fully meet PLOS ONE’s publication criteria as it currently stands. Therefore, we invite you to submit a revised version of the manuscript that addresses the points raised during the review process.

We would appreciate receiving your revised manuscript by Feb 24 2020 11:59PM. To enhance the reproducibility of your results, we recommend that if applicable you deposit your laboratory protocols in protocols.io, where a protocol can be assigned its own identifier (DOI) such that it can be cited independently in the future. For instructions see: http://journals.plos.org/plosone/s/submission-guidelines#loc-laboratory-protocols

We look forward to receiving your revised manuscript.

Kind regards,

Christopher Edward Cornwall, Ph.D.

Academic Editor

PLOS ONE

Reviewers' comments:

Reviewer's Responses to Questions

**Comments to the Author**

1. If the authors have adequately addressed your comments raised in a previous round of review and you feel that this manuscript is now acceptable for publication, you may indicate that here to bypass the “Comments to the Author” section, enter your conflict of interest statement in the “Confidential to Editor” section, and submit your "Accept" recommendation.

Reviewer #1: (No Response)

Reviewer #2: (No Response)

2. Is the manuscript technically sound, and do the data support the conclusions?

Reviewer #1: Yes

Reviewer #2: Yes

3. Has the statistical analysis been performed appropriately and rigorously? 

Reviewer #1: Yes

Reviewer #2: Yes

4. Have the authors made all data underlying the findings in their manuscript fully available?

Reviewer #1: Yes

Reviewer #2: Yes

5. Is the manuscript presented in an intelligible fashion and written in standard English?

Reviewer #1: Yes

Reviewer #2: Yes

6. Review Comments to the Author

Reviewer #1: Overall I am happy with the authors responses, but there are a few minor things I like to see better clarified

Title: I remain unconvinced that this criticism of the use of coccolith morphotypes as a proxy for coccolith mass applies to all global Ehux populations. The work only examined Canary Island material, and hence I suggest a cautious subtitle:

"Coccolith mass and morphology of different Emiliania huxleyi morphotypes: a critical examination of Canary Island material"

In the abstract, explain that CPR stands for Circular Polarizer Retardation

Final sentence of abstract: ..cannot be uniformly used as reliable indicators...

"line 362: estimated the same way

line 423: to obtain insights

line 594-95: and uncritically using coccolith morphology as a proxy...

Reviewer #2: The authors have thoroughly revised and improved the manuscript, including having clarified some of the methodologies, expanded the abstract and re-organized the discussion section. However, I still find some of the discussion hard to follow and need to be improved for more clarity.

The study mainly estimated the coccolith thickness and calcite mass of coccolithophore Emiliania huxleyi collected off the northwestern African coast by combining the CPR-method and MCMC analysis. The major results indicate that the coccolith length and thickness did not necessarily correlate with the characteristics of different E. huxleyi morphotypes. The calcite mass was then calculated based on the coccolith length and thickness, and the authors concluded that the morphotype appearance and relative abundance can not be used as reliable indicators of E. huxleyi calcification or calcite production.

This conclusion is contradicted with many of the previously published research. For this reason, the authors have added two paragraphs of “comparison with other studies” to explain the discrepancy between the methods used in the present study and other studies (polarized microscopic and SEM imaging methods). However, I still think the comparison is missing some important point. My major concern is that the present study mainly used the unified calcite density and simplified coccolith volume (equation 2) to generate the final conclusion, and thereby the effects of different morphotypes are ignored based on my understanding – how is this assumption valid compared to other methods? For example, Beaufort et al. (2008, biogeosciences) used the relative lightness of crystals as indicators of calcite mass.

7. PLOS authors have the option to publish the peer review history of their article (what does this mean?). If published, this will include your full peer review and any attached files.

Reviewer #1: No

Reviewer #2: No

---

## [Author Response · Author response to Decision Letter 1]

23 Feb 2020

Response to reviewers

Reviewer #1: Overall I am happy with the authors responses, but there are a few minor things I like to see better clarified

Title: I remain unconvinced that this criticism of the use of coccolith morphotypes as a proxy for coccolith mass applies to all global Ehux populations. The work only examined Canary Island material, and hence I suggest a cautious subtitle:

"Coccolith mass and morphology of different Emiliania huxleyi morphotypes: a critical examination of Canary Island material" 

Reply:

We changed the title as suggested by the reviewer.

"In the abstract, explain that CPR stands for Circular Polarizer Retardation

Final sentence of abstract: ..cannot be uniformly used as reliable indicators...

line 362: estimated the same way

line 423: to obtain insights

line 594-95: and uncritically using coccolith morphology as a proxy... "

Reply:

We changed the lines and abstract as suggested.

Reviewer #2: The authors have thoroughly revised and improved the manuscript, including having clarified some of the methodologies, expanded the abstract and re-organized the discussion section. However, I still find some of the discussion hard to follow and need to be improved for more clarity.

The study mainly estimated the coccolith thickness and calcite mass of coccolithophore Emiliania huxleyi collected off the northwestern African coast by combining the CPR-method and MCMC analysis. The major results indicate that the coccolith length and thickness did not necessarily correlate with the characteristics of different E. huxleyi morphotypes. The calcite mass was then calculated based on the coccolith length and thickness, and the authors concluded that the morphotype appearance and relative abundance can not be used as reliable indicators of E. huxleyi calcification or calcite production.

This conclusion is contradicted with many of the previously published research. For this reason, the authors have added two paragraphs of “comparison with other studies” to explain the discrepancy between the methods used in the present study and other studies (polarized microscopic and SEM imaging methods). However, I still think the comparison is missing some important point. My major concern is that the present study mainly used the unified calcite density and simplified coccolith volume (equation 2) to generate the final conclusion, and thereby the effects of different morphotypes are ignored based on my understanding – how is this assumption valid compared to other methods? For example, Beaufort et al. (2008, biogeosciences) used the relative lightness of crystals as indicators of calcite mass. 

Reply:

We agree with Reviewer Two that our results appear to contradict many previously published papers that utilised the method first published by Beaufort (2005). The contradictory results are not surprising because it has been demonstrated by several studies that the method published by Beaufort (2005) is flawed in many ways (see e.g. Bollmann (2013, 2014), Lochte (2014), Beaufort et al. 2014; Gonzalez-Lemos (2018). The method is not only inaccurate as it significantly overestimates coccolith weights, it also leads to irreproducible results because of a flawed calibration method (see publications above). 

Having said this, the method used in our research is based on the same basic principles as reported by Beaufort (2005). It uses “the relative lightness of crystals as indicators of calcite mass” and image analysis techniques (thresholding) to detect the outline/area of a coccolith and measure the grey value of individual pixels (lightness) within the area as a proxy for coccolith thickness/mass. 

Furthermore, using the average (unified according to Reviewer Two) grey values/lightness/thickness of a coccolith to infer the mass/weight of a certain length is also a technique that Beaufort has used in many publications (e.g. Beaufort et al. (2008), Biogeosciences; Beaufort et al. (2011), Nature). Here is one excerpt from Beaufort et al. (2008) Biogeosciences page 1105 where 4 pixel/0.6µm were added to the length of coccoliths and the average (unified according to Reviewer Two) grey value/lightness/thickness were extrapolated to calculate coccolith mass/weight:

Quote: “The Coccolith Analyser measures the grey level of objects, their diameter and surface, and tabulates the results. There is a bias of 0.6μm in the measurement of the diameter of small, and dim objects, such as coccoliths. This is because we apply a minimum Grey Level threshold above background. This threshold erodes 2 pixels of the periphery of dim objects. Each pixel is 0.15μm, and 4 pixels are eroded in total when the long diameter is measured. Thus we added 0.6μm to the measurement of coccolith length to compensate for this automatic loss. Another small bias in the measurements of the length of coccoliths of E. huxleyi exists, the distal shield being not detected in its entirety in cross-polarized light. There-fore the length of E. huxleyi presented here are slightly underestimated. We estimated this bias to a factor of 1.25 by comparing measures from the Coccolith analyzer and from SEM.”

Therefore, we do not agree with the following concern of Reviewer Two: 

Quote “My major concern is that the present study mainly used the unified calcite density and simplified coccolith volume (equation 2) to generate the final conclusion, and thereby the effects of different morphotypes are ignored based on my understanding – how is this assumption valid compared to other methods? For example, Beaufort et al. (2008, biogeosciences) used the relative lightness of crystals as indicators of calcite mass.”

In contrast to Beaufort et al. (2008), we do not extrapolate grey values/thickness to missing pixels/length. We use actual grey value/thickness measurements of coccoliths and use the average grey value of a coccolith in the LM to estimate the weight of similar sized coccoliths from SEM observations. We have not only overcome the size bias introduced by unsuitable thresholding techniques used by Beaufort (2005) by applying the Canny edge detection technique, we have also increased the taxonomic resolution significantly by using circular polarised light. In contrast to Beaufort et. al. (2008, 2011) who were only able to identify coccoliths at the family level, we were able to distinguish between species within the Noelaerhabdaceae. 

In conclusion, our data quality is significantly improved compared to any publication that has used the method by Beaufort (2005). Moreover, to the best of our knowledge, there is NO study that has been able or attempted to estimate the mass of different morphotypes at sub-species level on a light microscope. Therefore, it appears to be odd that Reviewer Two takes issue with our technique, results and interpretation with reference to other light microscope studies that were not even able to distinguish between species within the Noelaerhabdaceae.

---

## [Decision Letter · Decision Letter 2]

4 Mar 2020

Coccolith mass and morphology of different Emiliania huxleyi morphotypes: a critical examination using Canary Islands

material

PONE-D-19-23241R2

Dear Dr. Linge Johnsen,

We are pleased to inform you that your manuscript has been judged scientifically suitable for publication and will be formally accepted for publication once it complies with all outstanding technical requirements.

With kind regards,

Christopher Edward Cornwall, Ph.D.

Academic Editor

PLOS ONE

Additional Editor Comments (optional):

Reviewers' comments:

Reviewer's Responses to Questions

**Comments to the Author**

1. If the authors have adequately addressed your comments raised in a previous round of review and you feel that this manuscript is now acceptable for publication, you may indicate that here to bypass the “Comments to the Author” section, enter your conflict of interest statement in the “Confidential to Editor” section, and submit your "Accept" recommendation.

Reviewer #2: All comments have been addressed

2. Is the manuscript technically sound, and do the data support the conclusions?

Reviewer #2: (No Response)

3. Has the statistical analysis been performed appropriately and rigorously? 

Reviewer #2: (No Response)

4. Have the authors made all data underlying the findings in their manuscript fully available?

Reviewer #2: (No Response)

5. Is the manuscript presented in an intelligible fashion and written in standard English?

Reviewer #2: Yes

6. Review Comments to the Author

Reviewer #2: (No Response)

7. PLOS authors have the option to publish the peer review history of their article (what does this mean?). If published, this will include your full peer review and any attached files.

Reviewer #2: No

---

## [Editor Report · Acceptance letter]

9 Mar 2020

PONE-D-19-23241R2 

Coccolith mass and morphology of different *Emiliania huxleyi* morphotypes: a critical examination using Canary Islands
material 

Dear Dr. Linge Johnsen:

I am pleased to inform you that your manuscript has been deemed suitable for publication in PLOS ONE. Congratulations! Your manuscript is now with our production department. 

With kind regards,

on behalf of

Dr. Christopher Edward Cornwall 

Academic Editor

PLOS ONE